# Adversarial Training Helps Transfer Learning via Better Representations

**Zhun Deng***
Harvard University
Cambridge, MA 02138
zhundeng@g.harvard.edu

**Linjun Zhang***
Rutgers University
Piscataway, NJ 08854
linjun.zhang@rutgers.edu

**Kailas Vodrahalli**
Stanford University
Stanford, CA 94025
kailasv@stanford.edu

**Kenji Kawaguchi**
Harvard University
Cambridge, MA 02138
kkawaguchi@fas.harvard.edu

**James Zou**
Stanford University
Stanford, CA 94025
jamesz@stanford.edu

## Abstract

Transfer learning aims to leverage models pre-trained on source data to efficiently adapt to target setting, where only limited data are available for model fine-tuning. Recent works empirically demonstrate that adversarial training in the source data can improve the ability of models to transfer to new domains. However, why this happens is not known. In this paper, we provide a theoretical model to rigorously analyze how adversarial training helps transfer learning. We show that adversarial training in the source data generates provably better representations, so fine-tuning on top of this representation leads to a more accurate predictor of the target data. We further demonstrate both theoretically and empirically that semi-supervised learning in the source data can also improve transfer learning by similarly improving the representation. Moreover, performing adversarial training on top of semi-supervised learning can further improve transferability, suggesting that the two approaches have complementary benefits on representations. We support our theories with experiments on popular data sets and deep learning architectures.

## 1 Introduction

Transfer learning is a popular methodology to obtain well-performing machine learning models in settings where high-quality labeled data is scarce [20, 48]. The general idea of transfer learning to take a pre-trained model from a source domain—where labeled data is abundant—and adapt it to a new target domain. Because the target data distribution often differs from the source setting, standard transfer learning fine-tunes the model using a small-amount of labeled data from the target domain. In many applications, the fine-tuning is performed only on the last few layers of the network if the amount of target data is limited or if the one only has access to a representation (*i.e.* intermediate layers) produced by the model instead of the full model.

Transfer learning has demonstrated substantial empirical success and there is an exciting literature investigating different approaches to making transfer learning more effective [26, 27]. Recent experiments empirically demonstrated an intriguing phenomenon that models that are trained using adversarial-robust optimization on the source data transfer better to target data compared to non-adversarially trained models. We illustrate this phenomenon in Figure 1, which replicates the findings

---

*Equal contribution.

35th Conference on Neural Information Processing Systems (NeurIPS 2021).

in [46]. Here two models are trained on the full ImageNet and 10% of ImageNet using different levels of adversarial training—$\epsilon$ is the $l_2$ magnitude of the adversarial attack. Following [46], we fine-tuned the last layer of the models using data from CIFAR-10 and plot the final accuracy on the target CIFAR-10. Adversarial training ($\varepsilon > 0$) significantly improves the transfer performance compared to model without adversarial training ($\varepsilon = 0$). Additional experiments demonstrating this effect are provided in [46, 55], however it is still an open question how adversarial training in source helps transfer learning.

As our **first contribution**, we initialize the study of how adversarial training helps fixed-feature transfer learning from a theoretical perspective. Our analysis shows how that adversarial training on the source learns a better representation such that fine-tuning on this representation leads to better performance on the target. Interestingly, we show that the robust representation can help transfer learning even when the source performance declines due to adversarial training. To the best of our knowledge, this is the first rigorous analysis of the effect of adversarial training on transfer learning.

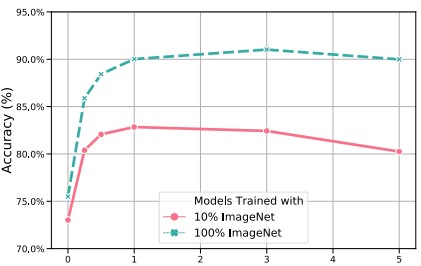

Figure 1: Transfer accuracy improves with adversarial training on source task. We plot target task (CIFAR-10) accuracy across different levels of $\ell_2$-adversarial training on the source task (ImageNet). The value of $\varepsilon$ corresponds to the size of the adversarial attack; *i.e.*, $\varepsilon = 0$ indicates no adversarial training. The two curves correspond to training the source model using all of ImageNet and a 10% subsample of ImageNet.

As our **second contribution**, we extend our analysis to show that semi-supervised learning using pseudo-labeling can similarly lead to better representations for transfer learning. We support our theory with empirical experiments. Moreover our experiments demonstrate for the first time that performing adversarial training on top of pseudo-labeling in the source can further boost transfer learning performance. This suggests that the two data augmentation techniques of adversarial training and pseudo-labeling have complementary benefits on learned representations.

As a **third technical contribution**, we generalize the techniques in prior papers for analyzing transfer learning in regressions to classification settings, where adversarial training and pseudo-labeling are more commonly used. Together, our results provide a useful and tractable framework to understand factors that improve transfer learning.

**Related Work**    Adversarial robust optimization has been a major focus in machine learning security [7, 16, 33, 23, 10, 39]. A serious of works has been proposed to increase the adversarial robustness both empirically [34, 37, 1] and theoretically [14, 30, 44, 32, 13, 29, 24, 18, 62]. Meanwhile, other works demonstrate how to quantify the trade-off between adversarial robustness and standard accuracy [61, 47, 11, 50, 17]. Recently [56, 46] empirically studied the transfer performance of adversarially robust networks, but it is still not clear yet why adversarial training leads to a better transfer from a theoretical perspective.

Transfer learning has been used in a variety of applications, ranging from medical imaging [43], natural language processing [25, 15], to object detection [31, 49]. On the theoretical side, some prior works [3, 5, 36, 58] studied the test accuracy of the target task in the multi-task learning setting. More recent work [53, 52, 21, 59] focused more on the representation learning and provide a theoretical framework to study linear representation in the *regression* setting. In this work, we provide a counterpart to theirs and studies the *classification* setting. Prior works in semi-supervised learning largely focus on improving the prediction accuracy with unlabeled data [66, 65, 6]. Works have also shown that semi-supervised learning can improve adversarial robustness [11, 19]. Several works have identified that using unlabeled data can empirically improve transfer learning [64, 63, 38], but a rigorous theoretical understanding of why this happens is lacking.

## 2 Preliminaries and model setup

**Notation.**    We use $[m]$ for $\{1, 2, \cdots, m\}$ for any $m \in \mathbb{N}^+$ and for any set $S$, let $|S|$ to denote the cardinality of $S$. For a matrix $M$, we denote $\sigma_k(M)$ as the $k$-th singular value of matrix $M$. We

use $\mathbb{O}_{m \times l}$ to denote the space of matrices of dimension $m \times l$ whose columns are orthonormal and use $\mathbb{S}^{p-1}$ to denote the unit sphere of dimension $p$. For two real matrices $E, F \in \mathbb{O}_{m \times l}$, we denote the subspaces spanned by the column vectors of $E$ and $F$ by $\mathcal{E}$ and $\mathcal{F}$ correspondingly. The subspace distance between $\mathcal{E}$ and $\mathcal{F}$ is defined as $\|\sin \Theta(E, F)\|_F$ [60], where $\Theta(E, F) = \text{diag}(\cos^{-1} \sigma_1(E^\top F), \cdots, \cos^{-1} \sigma_l(E^\top F))$. For a vector $v$, we use $\|v\|_q$ to denote the $\ell_q$ norm. Let $\lesssim$ and $\gtrsim$ denote "less than" and "greater than" up to a universal constant respectively. $a \ll b$ to denote $b \geq C \cdot a$ for a sufficiently large universal constant $C$. Our use of $O(\cdot), \Omega(\cdot), o(\cdot)$ follows the standard literature of computer science. With some abuse of notation, we also write $a = \Theta(b)$ if $a = O(b)$ and $a = \Omega(b)$ for $a, b \in \mathbb{R}$.

**Data generating processes.** We assume there are $T$ source tasks. For each task $t \in [T]$, we have corresponding training data set of size $n_t$, *i.e.* $S_t = \{(x_1^{(t)}, y_1^{(t)}), \cdots, (x_{n_t}^{(t)}, y_{n_t}^{(t)})\}$, where $x_i^{(t)} \in \mathcal{X} \subseteq \mathbb{R}^p$ and $y_i^{(t)} \in \{-1, 1\}$ are i.i.d. drawn from a joint distribution $\mathcal{P}_{x,y}^{(t)}$. We further denote $n = \min_{t \in [T]} n_t$. In other words, $n$ is the smallest size of source data sets. The goal of transfer learning is to learn from multiple source tasks in the hope of learning a common representation such that for a target task with distribution $\mathcal{P}_{x,y}^{(T+1)}$, we only need few data points to learn extra structures beyond the common representation and the learned model still achieves good prediction performance. With this spirit, we assume that for $t \in [T+1]$, $\{(x_i^{(t)}, y_i^{(t)})\}_i^{n_t}$ are i.i.d. drawn from $\mathcal{P}_{x,y}^{(t)}$, such that

$$x_i^{(t)} = \eta_i^{(t)} + y_i^{(t)} \mu_t, \tag{1}$$

for i.i.d. noise $\eta_i^{(t)}$ that is independent of $y_i^{(t)}$, where $\mu_t = Ba_t \in \mathbb{R}^p$, $a_t \in \mathbb{R}^r$ and $B \in \mathbb{R}^{p \times r}$ is an orthonormal matrix representing the projection onto a subspace, *i.e.* $B^\top B = I_r$. Here, $B$ is the common structure shared among all the source tasks and the target task, and $a_t$'s are task-specific parameters. Although this model is simple, the analysis is already highly nontrivial, and it captures the essense of the problem in transfer learning. In fact, similar models haves been considered in [52, 21]. Specifically, we consider the case where $r \ll p$. It can be viewed in a way that the data is generated by mapping low dimensional data signal to the high-dimension, which coincides with the fact that commonly used real image data sets lie in the lower dimensional manifolds. In addition, we assume the noise term $\eta_i^{(t)}$ is of zero-mean and is $\rho_t^2$-sub-gaussian, *i.e.* $\mathbb{E}[\exp(\lambda v^\top \eta_i^{(t)})] \leq \exp(\lambda^2 \rho_t^2/2)$ for all $v \in \mathbb{S}^{p-1}$ and $\lambda \in \mathbb{R}$. Throughout this paper, we consider $\rho_t = \Theta(1)$ for all $t \in [T]$.

**Remark 1.** *(i). The sub-gaussian assumption is quite flexible since many commonly used data sets such as image sets are all bounded, which implies sub-gaussianity. (ii). Different from the regression settings considered in previous theoretical work on transfer learning[21, 52], we focus on **classification settings**, in which adversarial training is more commonly studied.*

**Loss functions.** The loss functions considered in this paper take the following form: for each task $t \in [T+1]$,

$$\ell(x, y, w_2^{(t)}, W_1) = -yf^{(t)}(x), \tag{2}$$

where $f^{(t)}(x)$ is a two-layer linear neural network parametrized by $W_1$ and $w_2^{(t)}$, *i.e.* $f^{(t)}(x) = w_2^{(t)\top} W_1^\top x$, with $W_1 \in \mathbb{O}_{p \times r}$, $w_2^{(t)} \in \mathbb{R}^{r \times 1}$ and $\|w_2^{(t)}\| \leq 1$. Here, we mainly consider the case $W_1$ is well-specified, *i.e.* with the same dimension of $B$. Our argument can be further extended to the case where $r$ is unknown by first estimating $r$ and details are left to the appendix. We put norm constraint on $w_2^{(t)}$ since otherwise the minimizer is always of norm infinity. The loss function in (2) along with its variants have been commonly used in the theoretical machine learning community [47, 19]. Although in its simple form, it has been consistently useful to shed light upon complex phenomena. Meanwhile, even under this natural setting, it is highly non-trivial to demonstrate the effect of adversarial training in transfer learning.

Roughly speaking, like most settings in transfer learning, $W_1$ is assumed to be the common weights shared among the models for all source tasks so as to learn a "good" common representation. For each individual task $t$, parameter $w_2^{(t)}$ aims to perform task-specific linear classification. We leave the detailed discussions about how to take advantage of combining all the source tasks and obtaining a "good" $W_1$ to Section 3. Further, we denote the empirical loss for task $t$ as $\hat{L}(S_t, w_2^{(t)}, W_1) = \sum_{i=1}^{n_t} -y_i^{(t)} \langle W_1 w_2^{(t)}, x_i^{(t)} \rangle / n_t$. The expected loss for task $t$ is $L(\mathcal{P}_{x,y}^{(t)}, w_2^{(t)}, W_1) = -\mathbb{E}_{(x,y) \sim \mathcal{P}_{x,y}^{(t)}}[y \langle W_1 w_2^{(t)}, x \rangle]$.

**Problem Setup** In *fixed-representation* transfer learning, the first step is to learn the common representation in the model architectures using data from source tasks. The representation (e.g. the penultimate layer of a neural network) is then fixed. Finally, the target data is used to train or fine-tune a small model on top of the representation. Following this popular practice, in our model setting, we use the data of $T$ source tasks $\{S_t\}_{t=1}^{T}$ to obtain an estimator $\hat{W}_1$. Then, we use the data of target task $S_{T+1}$ to obtain an estimator $\hat{w}_2^{(T+1)}$ of the task-specific parameter. Our evaluation criteria is the *excess risk*:

$$\mathcal{R}(\hat{W}_1, \hat{w}_2^{(T+1)}) = L(\mathcal{P}_{x,y}^{(T+1)}, \hat{w}_2^{(T+1)}, \hat{W}_1) - \min_{\|w_2\| \leq 1, W_1 \in \mathbb{O}_{p \times r}} L(\mathcal{P}_{x,y}^{(T+1)}, w_2, W_1). \quad (3)$$

# 3 Adversarial Training Help Representation Learning

In this section, we demonstrate our results about how adversarial training can learn a better representation, and therefore leads to smaller excess risks. We first describe our algorithm, and demonstrate the near-optimality of our algorithm in representation learning by a minimax lower bound. We then demonstrate for the settings where data has varying noise-signal ratios or sparsity structures, how $\ell_2$ or $\ell_\infty$-adversarial training can help improve the representation learning.

## 3.1 Representation learning algorithm

Recall that the loss function for each task is $\ell(x, y, w_2^{(t)}, W_1) = -y w_2^{(t)\top} W_1^\top x$, where $W_1 \in \mathbb{O}_{p \times r}$ is a common structure in model architectures shared among all the source tasks and the target set. In the spirit of transfer learning, the goal is to jointly learn $W_1$ from source tasks and then use the data from the target task to learn its task-specific parameter $w_2^{(T+1)}$. Here, $W_1$ essentially aims to recover the common structure $B$ in the data generating processes Eq.(1) (or more rigorously, recover the column space of $B$), such that the obtained estimator $\hat{W}_1$ satisfies $\|\sin\Theta(\hat{W}_1, B)\|_F \to 0$.

Note that in our two-layer linear neural network structure, optimizing $w_2$ and $W_1$ simultaneously for a single task has the issue of non-identifiability – the loss value will not change if we multiply an orthonormal matrix $\Lambda \in \mathbb{R}^{r \times r}$ to $W_1$ and $\Lambda^{-1}$ to $w_2$. However, we still can jointly learn a good estimator $\hat{W}_1$ to recover $B$ following a similar method in [52] via singular value decomposition (SVD). In particular, we first simultaneously optimize $w_2^{(t)}$ and $W_1$ for each individual task for $t \in [T]$, which is equivalent to optimizing a single parameter $\beta_t = W_1^\top w_2^{(t)}$ (since $W_1$ is an orthonormal matrix, the norm of $\beta_t$ is still upper bounded by 1). Then, we apply SVD to the matrix consisting of the optimizers $\hat{\beta}_t$'s to obtain $\hat{W}_1$. In the final step, we use $S_{t+1}$ to learn $w_2^{(t+1)}$.

---

**Algorithm 1** Learning for Linear Representations

**Input:** $\{S_t\}_{t=1}^{T+1}$

Step 1: Optimize the loss function on each individual source task $t \in [T]$ and obtain

$$\hat{\beta}_t = \mathrm{argmin}_{\|\beta_t\| \leq 1} \frac{1}{n_t} \sum_{i=1}^{n_t} -y_i^{(t)} \langle \beta_t, x_i^{(t)} \rangle.$$

Step 2: $\hat{W}_1 \leftarrow$ top-$r$ SVD of $[\hat{\beta}_1, \hat{\beta}_2, \cdots, \hat{\beta}_T]$.
Step 3: $\hat{w}_2^{(T+1)} \leftarrow \mathrm{argmin}_{\|w_2^{(T+1)}\| \leq 1} \frac{1}{n_{T+1}} \sum_{i=1}^{n_{T+1}} -y_i^{(T+1)} \langle w_2^{(T+1)} \hat{W}_1, x_i^{(T+1)} \rangle.$
**Return** $\hat{W}_1, \hat{w}_2^{(T+1)}$.

---

Next, we provide a lemma about the representation learning in the two-layer linear neural network model under the assumption below. Combining this lemma with a minimax lower bound, we will show that adversarial training cannot have any gain in representation or transfer learning without extra special data structures, which motivates our subsequent theories. To facilitate the presentation, let us define $M = [a_1/\|a_1\|, a_2/\|a_2\|, \cdots, a_T/\|a_T\|]$.

**Assumption 1** (Task normalization and diversity). *For all the tasks, $\|a_t\| = \Theta(1)$ for all $t \in [T+1]$ and $\sigma_r(M^\top M/T) = \Omega(1/r)$.*

**Remark 2.** *Throughout the paper, we consider the low-rank case, where $r$ is smaller than $T$ and $p$. Meanwhile, notice that $\|M\|_F^2 = T = \sum_{i=1}^r \sigma_i^2(M)$, this assumption implies the condition number $\sigma_1(M)/\sigma_r(M) = O(1)$, which roughly means $\{a_i/\|a_i\|\}_{i=1}^T$ cover all the directions of $\mathbb{R}^r$ evenly.*

Loosely speaking, if we denote $\hat{\mu}_{T+1} = \sum_{i=1}^{n_{T+1}} x_i^{(T+1)} y_i^{(T+1)}/n_{T+1}$, under some regularity conditions, with high probability

$$\mathcal{R}(\hat{W}_1, \hat{w}_2^{(T+1)}) = L(\mathcal{P}_{x,y}^{(T+1)}, \hat{w}_2^{(T+1)}, \hat{W}_1) - \min_{\|w_2\|\leq 1, W_1 \in \mathbb{O}_{p\times r}} L(\mathcal{P}_{x,y}^{(T+1)}, w_2, W_1)$$

$$\lesssim \underbrace{\|\sin\Theta(\hat{W}_1, B)\|_F}_{\text{representation error}} + \underbrace{\|B^\top \hat{\mu}_{T+1} - B^\top \mu_{T+1}\|}_{\text{task-specific error}}. \tag{4}$$

The task-specific error is easy to deal with given Eq. (4), we mainly focus on providing a lemma to characterize the representation error.

**Lemma 1.** *Under Assumption 1, if $n > c_1 \max\{pr^2/T, r^2\log(1/\delta)/T, r^2\}$ for some universal constant $c_1 > 0$ and $2r \leq \min\{p, T\}$, for all $t \in [T]$. For $\hat{W}_1$ obtained in Algorithm 1, with probability at least $1 - O(n^{-100})$,*

$$\|\sin\Theta(\hat{W}_1, B)\|_F \lesssim r\left(\sqrt{\frac{1}{n}} + \sqrt{\frac{p}{nT}} + \sqrt{\frac{\log n}{nT}}\right).$$

Application of Lemma 1 gives us the following corollary about the excess risk $\mathcal{R}(\hat{W}_1, \hat{w}_2^{(T+1)})$.

**Corollary 1.** *Under Assumption 1, if $n > c_1 \max\{pr^2/T, r^2\log(1/\delta)/T, r^2, rn_{T+1}\}$ for some universal constant $c_1 > 0$, $2r \leq \min\{p, T\}$, then for $\hat{W}_1$ obtained in Algorithm 1, with probability at least $1 - O(n^{-100})$,*

$$\mathcal{R}(\hat{W}_1, \hat{w}_2^{(T+1)}) \lesssim \sqrt{\frac{r + \log n}{n_{T+1}}} + \sqrt{\frac{r^2 p}{nT}}.$$

**Remark 3.** *Lemma 1 and Corollary 1 provide counterparts of the bound of subspace distance and excess risk studied in [52, 21] under the setting of regression models. Since they use squared losses, our bounds are different from theirs by square roots. Squaring our bounds provide results with similar rates as those in previous work. If we do not use data from source tasks, we will obtain an excess risk bound of order $\sqrt{p/n_{T+1}}$ instead, which will be significantly larger than the one in Corollary 1 if $r + \log n \ll p$ and $pr^2 \ll nT$, which happens in our low rank situation with abundant source task data.*

Meanwhile, we provide the following minimax lower bound to justify the near-optimality of our algorithm in learning the representation in general cases.

**Proposition 1.** *Let us consider the parameter space $\Xi = \{A \in \mathbb{R}^{p\times r}, B \in \mathbb{R}^{p\times r} : \sigma_r(A^\top A/T) \gtrsim 1, B^\top B = I_r\}$. If $nT \gtrsim rp$, we then have*

$$\inf_{\hat{W}_1} \sup_{\Xi} \mathbb{E}\|\sin\Theta(B, \hat{W}_1)\|_F \gtrsim \sqrt{\frac{rp}{nT}}.$$

**Remark 4.** *For high dimensional data such that $p$ is much larger than $T$ and $\log n$, the lower bound in Proposition 1 matches the upper bound in Lemma 1 up to a factor $\sqrt{r}$. Since $r$ is considered as a small constant in our settings, we can see that in general cases when there is no additional structural assumptions, our algorithm already obtains the near-optimal rate in representation learning. However, in later sections, when we introduce some additional structural assumptions such as varying signal-to-noise ratios and sparsity structures among tasks, which commonly happens in real applications, we will show that adversarial training can improve representation learning and further leads to smaller excess risks.*

### 3.2 How $\ell_2$-adversarial training improves representation learning for transfer

In this subsection, we consider the benefit of $\ell_2$-adversarial training. Specifically, if the signal-to-noise ratios varies among tasks in the sense that $\|a_t\|$'s have different scales, $\ell_2$-adversarial training can

lead to a sharper representation estimation error than standard training. In contrast, Lemma 1 and Proposition 1 demonstrate that under the case of uniform signal-to-noise ratios, adversarial training cannot have any gain over standard training. From a high-level perspective, signal-to-noise ratios determiine the difficulties of classification. For those tasks with small signal-to-noise ratios, while adversarial attacks make them even harder to perform classification (increase bias), but also make these tasks less competitive (decrease variance). Thus, adversarial training will bias the model to focus on learning the representation out of those with large signal-to-noise ratios.

**Assumption 2** (Varying signal-to-noise ratios). *For the $T$ source tasks, they can be divided into two disjoint sets. The first set is $S_1 = \{t \in [T] : \|a_t\| = \Theta(1)\}$, and the second set is $S_2 = \{t \in [T] : \|a_t\| = \Omega(\alpha_T)\}$, where $\alpha_T \to \infty$ as $T \to \infty$, and $S_1 \cup S_2 = [T]$. In addition, $|S_2|/T = \Theta(1)$.*

For the matrix $M = [a_1/\|a_1\|, a_2/\|a_2\|, \cdots, a_T/\|a_T\|]$, we further denote $M_{S_1}$ as the sub-matrix of $M$, whose columns consist of of $a_t/\|a_t\|$ for $t \in S_1$. For instance, if $S_1 = \{1, 2, 3\}$. then $M_{S_1} = [a_1/\|a_1\|, a_2/\|a_2\|, a_3/\|a_3\|]$. We define $M_{S_2}$ similarly.

**Assumption 3** (Task diversity). *For the $T$ source tasks, $\min\{\sigma_r(M_{S_2}^\top M_{S_2}/T), \sigma_r(M^\top M/T)\} = \Omega(1/r)$.*

**Remark 5.** *Assumption 2 indicates if we have more source tasks (larger $T$), more tasks with large signal-to-noise ratios would show up. Similar to Assumption 1, Assumption 3 requires both the columns in $M$ and $M_{S_2}$ cover $\mathbb{R}^r$ evenly.*

Now, we consider the adversarial training algorithm for $\ell_q$-attack for $q = 2, \infty$.

---

**Algorithm 2** Adversarial Learning for Linear Features

---

**Input:** $\{S_t\}_{t=1}^{T+1}, q$

Step 1: Optimize the adversarial loss function on each individual source task $t \in [T]$ and obtain

$$\hat{\beta}_t^{adv} = \text{argmin}_{\|\beta_t\| \le 1} \max_{\|\delta_i\|_q \le \varepsilon} \frac{1}{n_t} \sum_{i=1}^{n_t} -y_i^{(t)} \langle \beta_t, x_i^{(t)} + \delta_i \rangle.$$

Step 2: $\hat{W}_1^{adv} \leftarrow$ top-$r$ SVD of $[\hat{\beta}_1^{adv}, \hat{\beta}_2^{adv}, \cdots, \hat{\beta}_T^{adv}]$.
Step 3: $\hat{w}_2^{adv,(T+1)} \leftarrow \text{argmin}_{\|w_2^{(T+1)}\| \le 1} \frac{1}{n_{T+1}} \sum_{i=1}^{n_{T+1}} -y_i^{(T+1)} \langle w_2^{(T+1)} \hat{W}_1^{adv}, x_i^{(T+1)} \rangle.$
**Return** $\hat{W}_1^{adv}, \hat{w}_2^{adv,(T+1)}$.

---

The following theorem shows that even when the $\hat{\beta}_t^{adv}$'s obtained by $\ell_2$-adversarial training have large excess risk for each source task, the $\hat{W}_1^{adv}$ extracted from $[\hat{\beta}_1^{adv}, \hat{\beta}_2^{adv}, \cdots, \hat{\beta}_T^{adv}]$ can transfer knowledge from multiple source tasks better and result in a smaller excess risk on the target task.

**Theorem 1.** *Under Assumption 2 and 3, for $\|a_{T+1}\| = \alpha = \Omega(1)$, if $n > c_1 \max\{r^2, r/\alpha_T\} \cdot \max\{p \log T, \log n/T, 1\}$ and $n > c_2(\alpha\alpha_T)^2 r n_{T+1}$ for universal constants $c_1, c_2$, $2r \le \min\{p, T\}$. There exists a universal constant $c_3$, such that if we choose $\varepsilon \in [\max_{t \in S_1} \|a_t\| + c_3\sqrt{p \log T/n}, \min_{t \in S_2} \|a_t\| - c_3\sqrt{p \log T/n}]$ (this set will not be empty if $T, n$ are large enough), for $\hat{W}_1^{adv}, \hat{w}_2^{adv,(T+1)}$ obtained in Algorithm 2 with $q = 2$, with probability at least $1 - O(n^{-100})$,*

$$\|\sin\Theta(\hat{W}_1^{adv}, B)\|_F \lesssim (\alpha_T)^{-1} \left( \sqrt{\frac{r^2}{n}} + \sqrt{\frac{pr^2}{nT}} + \sqrt{\frac{r^2 \log n}{nT}} \right),$$

*and the excess risk*

$$\mathcal{R}(\hat{W}_1^{adv}, \hat{w}_2^{adv,(T+1)}) \lesssim \alpha\sqrt{\frac{r + \log n}{n_{T+1}}} + (\alpha_T)^{-1} \left( \sqrt{\frac{r^2 p}{nT}} \right).$$

Similar to Assumption 2, here $\alpha$ can also be a function of the target task data size $n_{T+1}$.

$\ell_2$**-adversarial training v.s. standard training:** Under the exact same conditions in Theorem 1, a simple modification of Lemma 1 leads to $\|\sin\Theta(\hat{W}_1, B)\|_F \lesssim \sqrt{r^2/n} + \sqrt{pr^2/(nT)} + \sqrt{r^2 \log n/(nT)}$ and $\mathcal{R}(\hat{W}_1, \hat{w}_2^{(T+1)}) \lesssim \alpha\sqrt{(r + \log n)/n_{T+1}} + \sqrt{r^2 p/(nT)}$ with high probability.

We can see that the adversarial training would lead to a better representation and an improved excess risk when $\alpha_T$ is growing. Such a scenario happens when the source data consist of a large diversity of tasks with varying difficulties of classification. Our proof indeed reveals that adversarial training would help the model to focus on learning the representation from easy-to-classify tasks, and therefore improves the convergence rate of representation learning. The gain in representation learning further leads to smaller rates of $\mathcal{R}(\hat{W}_1^{adv}, \hat{w}_2^{adv,(T+1)})$ compared with $\mathcal{R}(\hat{W}_1, \hat{w}_2^{(T+1)})$.

### 3.3 How $\ell_\infty$-adversarial training improves representation learning for transfer

In this subsection, we further consider the benefit of $\ell_\infty$-adversarial training. It is well-recognized that commonly used real data sets, such as MNIST and CIFAR-10, actually lie in lower dimensional manifolds compared with their ambient dimensions. After certain transformations [2, 9], it is equivalent to having sparsity structure in the coordinates. We demonstrate that if there are some underlying sparsity structures in the mean parameters $\mu_t = Ba_t$ for $t \in [T]$, then $\ell_\infty$-adversarial training leads to sharper bounds regarding the representation error and excess risk. To facilitate the discussion, let us use $\mu_{t,j}$ to denote the $j$-th coordinates of $\mu_t$.

**Assumption 4** (Structural sparsity). *For an integer $s$ such that $0 < s < p$, we assume for all $t \in [T]$, $\operatorname{sign}(\mu_{t,j})$ are i.i.d. and $\mathbb{P}(\operatorname{sign}(\mu_{t,j}) = 0) = 1 - \eta_s$, $\mathbb{P}(\operatorname{sign}(\mu_{t,j}) = 1) = \mathbb{P}(\operatorname{sign}(\mu_{t,j}) = -1) = \eta_s/2$. We also refer $s$ as the sparsity level.*

Assumption 4 guarantees that the sparsity of each column is upper bounded by $O(s \log T)$ with high probability. Similar assumptions have been commonly used in the high-dimensional statistics literature [4, 51]. For $\ell_\infty$-adversarial training, we provide bounds obtained through adversarial training below.

**Theorem 2.** *Under Assumptions 1 and 4, if $n > c_1 \cdot r^2 \max\{s^2 \log^2 T/T, rn_{T+1}, 1\}$ for some universal constants $c_1 > 0$, $2r \le \min\{p, T\}$. There exists a universal constant $c_2$, such that if we choose $\varepsilon > c_2\sqrt{\log p/n}$, for and $\hat{W}_1^{adv}$, $\hat{w}_2^{adv,(T+1)}$ obtained in Algorithm 2 with $q = \infty$, with probability at least $1 - O(n^{-100}) - O(T^{-100})$,*

$$\|\sin\Theta(\hat{W}_1^{adv}, B)\|_F \lesssim r\left(\sqrt{\frac{1}{n}} + \sqrt{\frac{s^2}{nT}}\right) \cdot \log(T + p),$$

*and the excess risk*

$$\mathcal{R}(\hat{W}_1^{adv}, \hat{w}_2^{adv,(T+1)}) \lesssim \left(\sqrt{\frac{r + \log n}{n_{T+1}}} + r\sqrt{\frac{s^2}{nT}}\right) \cdot \log(T + p). \tag{5}$$

$\ell_\infty$**-adversarial training v.s. standard training:** Under the exact same conditions in Theorem 2, again, a simple modification of Lemma 1 shows that without adversarial training, with high probability, we have $\|\sin\Theta(\hat{W}_1, B)\|_F \lesssim r(\sqrt{1/n} + \sqrt{p/nT} + \sqrt{\log n/nT})$ and the excess risk $\mathcal{R}(\hat{W}_1, \hat{w}_2^{(T+1)}) \lesssim \sqrt{(r + \log n)/n_{T+1}} + r\sqrt{p/nT}$. Theorem 2 shows that $\ell_\infty$-adversarial training is able to learn significantly better representations when $s^2 \ll p$. This scenario is common in image classification where the label of an image only depends on a small set of feature. Our proof reveals that $\ell_\infty$-adversarial training would help remove the redundant features in the classification tasks and therefore improves the representation learning and the subsequent downstream prediction on target domain.

## 4 Pseudo-Labeling and Adversarial Training

In the previous section, we have shown that combining abundant data from source tasks with robust training can help learn a good classifier for the target task. Sometimes, however, even the sources have limited labeled data. In that case, data augmentation by incorporating unlabeled source data, which are easier to obtain, can be a powerful way to improve prediction accuracy. One of the most commonly used semi-supervised learning algorithms is the pseudo-labeling algorithm [12]. In this section, we explore how using pseudo-labeling in the source data can improve transfer learning and how adversarial training can further boost that improvement, both empirically and theoretically.

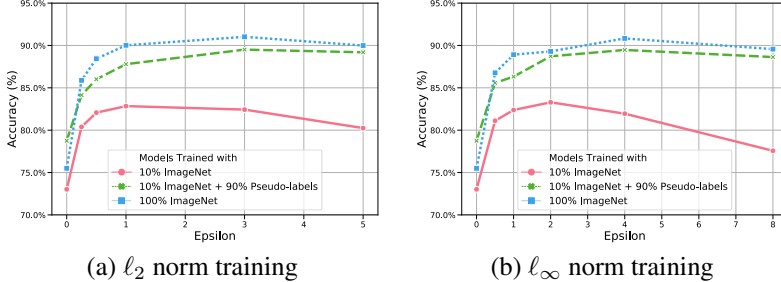

|                        |                        |
|:----------------------:|:----------------------:|
| (a) $\ell_2$ norm training | (b) $\ell_\infty$ norm training |

Figure 2: Comparison of target task (CIFAR-10) accuracy for models trained on source task (ImageNet) using (i). a 10% sample of data from the source task, (ii). the 10% sample with ground truth labels, and the remaining 90% with pseudo-labels, and (iii). 100% of the source task data with ground truth labels. Models trained on the source task with (a). $\ell_2$-adversarial training and (b). $\ell_\infty$-adversarial training both exhibit similar behavior. The x-axis refers to the magnitude, $\varepsilon$, used in adversarial training—larger values indicate allowing more difficult adversarial examples; 0 corresponds to no adversarial training. The $\varepsilon$ value in (b) is scaled up by 255 so it corresponds to pixel difference on a [0,255] scale.

**Experiments.** We perform empirical study of image classification. Our source tasks are image classification on ImageNet [45]; our target tasks are image classification on CIFAR-10 [28]. To simulate the pseudo-labeling setup, we sample 10% of ImageNet, train a ResNet-18 model on this sample (without adversarial training), and generate pseudo-labels for the remaining 90%. We then train a new source model using all of the source labeled and pseudo-labeled data with and without adversarial training. We use a public library for adversarial training [22]. The high-level approach for adversarial training is as follows: at each iteration, take a small number of gradient steps to generate adversarial examples from an input batch; then update network weights using the loss gradients from the adversarial batch.

In Figure 2, we plot the target task accuracy of models trained on our source task in 3 different settings, across different levels of adversarial training. Models in Figure 2 (a) and (b) are trained on the source task with $l_2$ and $l_\infty$-adversarial training respectively. We compare models trained on the source task using: 1) a fixed 10% sample of ImageNet,; 2) the 10% sample with ground truth labels and the remaining 90% sample using the generated pseudo-labels; and 3) all of ImageNet with its ground truth labels. Adversarial training boosts transfer performance in all 3 settings. Pseudo-labels also boost transfer performance. In the $\varepsilon = 0$ setting (i.e. no adversarial training), the highest target accuracy is obtained by using labeled examples with pseudo-labels (green points). Moreover, adversarial training with pseudo-labels also increases performance; at the optimal setting for adversarial training, the difference from using pseudo-labels instead of ground truth labels is only $1.5\%$.

Table 1: Effect of amount of pseudo-labels on transfer task performance (measured with accuracy). At $0\%$, we just use $10\%$ of data from the source task; at $900\%$, we use all remaining $90\%$ of data with pseudo-labels (this is 9 times the train set size). Adversarial training corresponds to using $\ell_2$-adversarial training with $\varepsilon = 1$ on the source task. Results on additional datasets in Appendix.

| Source Task | Target Task | +0% Pseudo-labels | +200% Pseudo-labels | +500% Pseudo-labels | +900% Pseudo-labels |
|---|---|---|---|---|---|
| ImageNet | CIFAR-10 | 73.0% | 73.8% | 77.1 % | 78.8 % |
| ImageNet (w/adv.training) | CIFAR-10 | 82.8% | 85.7% | 87.5 % | 87.8 % |
| ImageNet | CIFAR-100 | 51.0% | 52.9% | 55.3 % | 58.4% |
| ImageNet (w/adv.training) | CIFAR-100 | 62.6% | 65.2% | 68.1 % | 69.5 % |

In Table 1 we investigate how the amount of pseudo-labeled data affects performance. We train models in 2 settings: with adversarial and non-adversarial (standard) training on the source task. The adversarial training corresponds to $\ell_2$ norm adversarial training with $\varepsilon = 1$. Across all settings, we observe that robust training improves performance, and adding more pseudo-labeled data improves performance with diminishing returns.

**Theoretical illustration.** We further support the above experimental observations with theories. We denote the unlabeled input data for each source task $t \in [T]$ as $X_t^u = \{x_i^{u,(t)}\}_{i=1}^{n_t^u}$. The algorithm we analyze is as the following:

---
**Algorithm 3** Natural and Adversarial Learning for Linear Features with Pseudo-labeling

---
**Input:** $\{S_t\}_{t=1}^{T+1}, \{X_t^u\}_{t=1}^T, q$

Step 1: Train an initial classifier: $w_{init}^{(t)} = \text{argmin}_{\|w\| \leq 1} \frac{1}{n_t} \sum_{i=1}^{n_t} -y_i^{(t)} \langle w, x_i^{(t)} \rangle$

Step 2: Obtain pseudo labels: $y_i^{u,(t)} = sgn(\langle w_{init}^{(t)}, x_i^{(t)} \rangle)$

Step 3: Obtain augmented data sets $S_{t,aug}$ by combining $S_t$ and $\{(x_i^{u,(t)}, y_i^{u,(t)})\}_{i=1}^{n_t^u}$

Step 4: $(\hat{W}_{1,aug}, \hat{w}_{2,aug}^{(T+1)}) \leftarrow$ Algorithm 1$(S_{t,aug}, S_{T+1})$,

$\quad\quad (\hat{W}_{1,aug}^{adv} \hat{w}_{2,aug}^{adv,(T+1)}) \leftarrow$ Algorithm 2$(S_{t,aug}, S_{T+1}, q)$

**Return** $\hat{W}_{1,aug}, \hat{w}_{2,aug}^{(T+1)}, \hat{W}_{1,aug}^{adv} \hat{w}_{2,aug}^{adv,(T+1)}$

---

**Theorem 3.** *Denote $\tilde{n} = \min_{t \in [T]} n_t^u$ and assume $\tilde{n} > c_1 \max\{pr^2/T, r^2 \log(1/\delta)/T, r^2, n\}$ for some constant $c_1 > 0$. Assume $\sigma_r(M^\top M/T) = \Omega(1/r)$ and $n^{c_2} \gtrsim \tilde{n} \gtrsim n$ for some $c_2 > 1$, if $n \gtrsim (T + d)$ and $\min_{t \in [T]} \|a_t\| = \Theta(\log^2 n)$ and $\eta_i^{(t)} \sim \mathcal{N}_p(0, \rho_t^2 I^2)$ for $\rho_t = \Theta(1)$. Let $\hat{W}_{1,aug}$ obtained in Algorithm 3, with probability $1 - O(n^{-100})$,*

$$\|\sin\Theta(\hat{W}_{1,aug}, B)\|_F \lesssim r \left( \sqrt{\frac{1}{\tilde{n}}} + \sqrt{\frac{p}{\tilde{n}T}} + \sqrt{\frac{\log n}{\tilde{n}T}} \right).$$

Comparing the results above with Lemma 1, we theoretically justify that by incorporating unlabeled data, we are able to learn a better representation when $\tilde{n} \gg n$. In the following, we show that adversarial training, together with the pseudo-labeling, can further boost this improvement.

**Theorem 4.** *Under the same conditions as those in Theorem 3,*

*(a). For $\ell_2$ attack, under assumptions same to the those in Theorem 1, and additionally $\tilde{n} > c_1 \max\{r^2, r/\alpha_T\} \max\{p \log T, \log n/T, 1\}$ for a universal constant $c_1$, and choose $\varepsilon \in [\max_{t \in S_1} \|a_t\| + c_3 \sqrt{p \log T/\tilde{n}}, \min_{t \in S_2} \|a_t\| - c_3 \sqrt{p \log T/\tilde{n}}]$, we then have with probability at least $1 - O(n^{-100})$,*

$$\|\sin\Theta(\hat{W}_{1,aug}^{adv}, B)\|_F \lesssim (\alpha_T)^{-1} \left( \sqrt{\frac{r^2}{\tilde{n}}} + \sqrt{\frac{pr^2}{\tilde{n}T}} + \sqrt{\frac{r^2 \log(n)}{\tilde{n}T}} \right); \quad\quad (6)$$

*(b). For $\ell_\infty$ attack, under assumptions same to the those in Theorem 2, and additionally $\tilde{n} > C_1 \cdot r^2 \max\{s^2 \log^2 T/T, 1\}$ for a universal constant $C_1$. There exists a universal constant $c_2$, such that if we choose $\varepsilon > c_3 \sqrt{\log p/\tilde{n}}$, with probability at least $1 - O(n^{-100}) - O(T^{-100})$,*

$$\|\sin\Theta(\hat{W}_{1,aug}^{adv}, B)\|_F \lesssim r(\sqrt{\frac{1}{\tilde{n}}} + \sqrt{\frac{s^2}{\tilde{n}T}}) \cdot \log(T + p).$$

Similar to the interpretations of Theorems 1 and 2, Theorem 4 suggests that adversarial training can boost the representation learning either (i). when the signal to noise ratio is varying ($\ell_2$ adversarial training helps in this case) and (ii). where there are many redundant features in classification ($\ell_\infty$ adversarial training helps in this case). Same to the analysis before, we can obtain similar upper bounds on the excess risks as those in Theorems 1 and 2 by using Eq. (4).

## 5 Discussion

In this paper, we provide the first theoretical framework to explain how adversarial training on the source data improves transfer learning. We show that adversarial training helps learning a more robust representation, and therefore boosts the predictive performance on the target task. Additionally, we extend our analysis to the semi-supervised setting and show that adversarial training, together with pseudo-labeling, have complementary benefits and can further improve the transfer.

**Societal impacts and limitations** Transfer learning helps learn a well-performed machine learning model with only a small amount of labeled data from the target task. Our work contributes to this field by providing insights into factors that improve transfer learning. A limitation of our work is that we have to make some standard assumptions on the data generative distribution when developing theories, which were also made in several other theory papers. While the model is simple, it captures the essence of the problem studied in the paper and is the first tractable framework to study how adversarial training helps fixed-feature transfer learning. The analysis here are already challenging and are supported by our experiments.

# 6 Acknowledgements

The research of Linjun Zhang is supported by NSF DMS-2015378. The research of James Zou is supported by NSF CCF 1763191, NSF CAREER 1942926 and grants from the Silicon Valley Foundation and the Chan-Zuckerberg Initiative. This work is also in part supported by NSF award 1763665.

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
