# Appendix

**Outline**  We provide detailed proofs for all of our theories in Secs. A to F. Sec. G provides multiple additional experiments demonstrating that pseudo-labeling improves transfer learning and that combining pseudo-labeling with adversarial training in the source further improves tranferability. Sec. H provides additional details about our experiments.

Recall that in the main context, in Algorithm 1, we have $\hat{W}_1 \leftarrow$ top-$r$ SVD of $[\hat{\beta}_1, \hat{\beta}_2, \cdots, \hat{\beta}_T]$. Specifically, we assign the columns of $\hat{W}_1$ as the collection of the top-$r$ left singular vectors of $[\hat{\beta}_1, \hat{\beta}_2, \cdots, \hat{\beta}_T]$.

The rest of proofs are based on the above methodology.

## A  Proof of Lemma 1

Let us define $\hat{\mu}_t = \sum_{i=1}^{n_t} x_i^{(t)} y^{(t)}/n_t$ and $\mu_t = Ba_t$ for all $t \in [T+1]$.

Notice that

$$\hat{J} = (\hat{\mu}_1/\|\hat{\mu}_1\|, \cdots, \hat{\mu}_T/\|\hat{\mu}_T\|) = (\hat{\mu}_1, \cdots, \hat{\mu}_T)\text{diag}(\|\hat{\mu}_1\|^{-1}, \cdots, \|\hat{\mu}_T\|^{-1})$$

As a result, doing SVD for $\hat{J}$ to obtain left singular vectors is equivalent to doing SVD for $\hat{\Phi} = (\hat{\mu}_1, \cdots, \hat{\mu}_T)$ to obtain left singular vectors (up to an orthogonal matrix, meaning rotation of the space spanned by the singular vectors) since multiplying a diagonal matrix on the right does not affect the collection of left singular vectors. It further means doing SVD for $\hat{J}$ to obtain left singular vectors is equivalent to obtaining left singular vectors for $\hat{\Phi} = (\hat{\mu}_1, \cdots, \hat{\mu}_T)\text{diag}(\|\mu_1\|^{-1}, \cdots, \|\mu_T\|^{-1})$ (up to an orthogonal matrix).

We mainly adopt the Davis-Kahan Theorem in [60]. We further denote $\Phi = (\mu_1, \cdots, \mu_T)\text{diag}(\|\mu_1\|^{-1}, \cdots, \|\mu_T\|^{-1})$.

**Lemma 2** (A variant of Davis–Kahan Theorem). *Assume $\min\{T, p\} > r$. For simplicity, we denote $\hat{\sigma}_1 \geq \hat{\sigma}_2 \geq \cdots \geq \hat{\sigma}_r$ as the top largest $r$ singular value of $\hat{\Phi}$ and $\sigma_1 \geq \sigma_2 \geq \cdots \geq \sigma_r$ as the top largest $r$ singular value of $\Phi$. Let $V = (v_1, \cdots, v_r)$ be the orthonormal matrix consists of left singular vectors corresponding to $\{\sigma_i\}_{i=1}^r$ and $\hat{V} = (\hat{v}_1, \cdots, \hat{v}_r)$ be the orthonormal matrix consists of left singular vectors corresponding to $\{\hat{\sigma}_i\}_{i=1}^r$. Then,*

$$\|\sin\Theta(\hat{V}, V)\|_F \lesssim \frac{(2\sigma_1 + \|\hat{\Phi} - \Phi^*\|_{op})\min\{r^{0.5}\|\hat{\Phi} - \Phi^*\|_{op}, \|\hat{\Phi} - \Phi^*\|_F\}}{\sigma_r^2}.$$

*Moreover, there exists an orthogonal matrix $\hat{O} \in \mathbb{R}^{r \times r}$, such that $\|\hat{V}\hat{O} - V\|_F \leq \sqrt{2}\|\sin\Theta(\hat{V}, V)\|_F$, and*

$$\|\hat{V}\hat{O} - V\|_F \lesssim \frac{(2\sigma_1 + \|\hat{\Phi} - \Phi^*\|_{op})\min\{r^{0.5}\|\hat{\Phi} - \Phi^*\|_{op}, \|\hat{\Phi} - \Phi^*\|_F\}}{\sigma_r^2}.$$

It is worth noticing that actually $B$ plays the exact same role as $V$. Since $B$ has orthonormal columns, for $\phi$ we have

$$\Phi = B(a_1, \cdots, a_T)\text{diag}(\|\mu_1\|^{-1}, \cdots, \|\mu_T\|^{-1})$$
$$= B(a_1, \cdots, a_T)\text{diag}(\|a_1\|^{-1}, \cdots, \|a_T\|^{-1}).$$

Thus, $B$ is a solution of the SVD step in Algorithm 1.

**Lemma 3** (**Restatement of Lemma 1**). *Under Assumption 1, if $n > c_1 \max\{pr^2/T, r^2\log(1/\delta)/T, r^2\}$ for some universal constant $c_1 > 0$ and $2r \leq \min\{p, T\}$, for all $t \in [T]$. For $\hat{W}_1$ obtained in Algorithm 1, with probability at least $1 - O(n^{-100})$,*

$$\|\sin\Theta(\hat{W}_1, B)\|_F \lesssim r\left(\sqrt{\frac{1}{n}} + \sqrt{\frac{p}{nT}} + \sqrt{\frac{\log n}{nT}}\right).$$

*Proof.* By a direct application of Lemma 2, we can obtain

$$\|\sin\Theta(\hat{W}_1, B)\|_F \lesssim \frac{(2\sigma_1 + \|\hat{\Phi} - \Phi\|_{op})\min\{r^{0.5}\|\hat{\Phi} - \Phi\|_{op}, \|\hat{\Phi} - \Phi\|_F\}}{\sigma_r^2}$$

Besides, we know that the left singular vectors of $\Phi$ are the same as the ones of $M = [a_1, \cdots, a_T]$ since $\Phi = BM\mathrm{diag}(\|a_1\|^{-1}, \cdots, \|a_T\|^{-1})$.

To estimate $\|\hat{\Phi} - \Phi\|_{op} = \sup_{v \in \mathbb{S}^{p-1}} \|v^\top(\hat{\Phi} - \Phi)\|$, for any fixed $v \in \mathbb{S}^{p-1}$, by standard chaining argument in Chapter 6 in [57], we know that

$$\mathbb{P}\left(\|v^\top(\hat{\Phi} - \Phi)\| \gtrsim \sqrt{\frac{T}{n}} + \sqrt{\frac{\log(1/\delta)}{n}}\right) \leq \delta$$

Then, we use chaining again for the $\psi_2$-process $\{v : \|v^\top(\hat{\Phi} - \Phi)\|\}$, we obtain

$$\mathbb{P}\left(\sup_{v \in \mathbb{S}^{p-1}} \|v^\top(\hat{\Phi} - \Phi)\| \gtrsim \sqrt{\frac{p}{n}} + \sqrt{\frac{T}{n}} + \sqrt{\frac{\log(1/\delta)}{n}}\right) \leq \delta.$$

Besides, we know $\sigma_r(M) = \sqrt{T/r}$ by assumption 1, and we also have $\sum_{i=1}^r \sigma^2(M) = T$, thus, we know that $\sigma_1(M)$ and $\sigma_r(M)$ are both of order $\Theta(\sqrt{T/r})$

$$\|\sin\Theta(\hat{W}_1, B)\|_F \lesssim \frac{(\sqrt{\frac{p}{n}} + \sqrt{\frac{T}{n}} + \sqrt{\frac{\log(1/\delta)}{n}} + \sqrt{T/r})\sqrt{r}(\sqrt{\frac{p}{n}} + \sqrt{\frac{T}{n}} + \sqrt{\frac{\log(1/\delta)}{n}})}{T/r},$$

by simple calculation, we further have

$$\|\sin\Theta(\hat{W}_1, B)\|_F \lesssim r\sqrt{r}(\frac{1}{n} + \frac{p}{nT} + \frac{\log(1/\delta)}{nT}) + r(\sqrt{\frac{1}{n}} + \sqrt{\frac{p}{nT}} + \sqrt{\frac{\log(1/\delta)}{nT}}).$$

If we further have $n > r\max\{p/T, \log(1/\delta)/T, 1\}$, we further have

$$\|\sin\Theta(\hat{W}_1, B)\|_F \lesssim r(\sqrt{\frac{1}{n}} + \sqrt{\frac{p}{nT}} + \sqrt{\frac{\log(1/\delta)}{nT}}).$$

Plugging into $\delta = n^{-100}$, the proof is complete.

$\square$

## B    Proof of Corollary 1

**Corollary 2 (Restatement of Corollary 1).** *Under Assumption 1, if $n > c_1\max\{pr^2/T, r^2\log(1/\delta)/T, r^2, rn_{T+1}\}$ for some universal constant $c_1 > 0$, $2r \leq \min\{p, T\}$, then for $\hat{W}_1$ obtained in Algorithm 1, with probability at least $1 - O(n^{-100})$,*

$$\mathcal{R}(\hat{W}_1, \hat{w}_2^{(T+1)}) \lesssim \sqrt{\frac{r + \log n}{n_{T+1}}} + \sqrt{\frac{r^2 p}{nT}}.$$

*Proof.* By DK-lemma, we know there exists a $W_1^*$ such that $W_1^* \in \mathrm{argmin}_{W \in \mathbb{O}_{p \times r}} \|W^\top \mu_{T+1}\|$ (the minimizer is not unique, so we use $\in$ instead of $=$ to indicate $W_1^*$ belongs to the set consists of minimizers) and $\|W_1^* - \hat{W}_1\|$ is small.

$$\mathcal{R}(\hat{W}_1, \hat{w}_2^{(T+1)}) = L(\mathcal{P}_{x,y}^{(T+1)}, \hat{w}_2^{(T+1)}, \hat{W}_1) - \min_{\|w_2\| \le 1, W_1 \in \mathbb{O}_{p \times r}} L(\mathcal{P}_{x,y}^{(T+1)}, w_2, W_1)$$

$$= -\langle \frac{\hat{W}_1^\top \hat{\mu}_{T+1}}{\|\hat{W}_1^\top \hat{\mu}_{T+1}\|}, \hat{W}_1^\top \mu_{T+1} \rangle + \|W_1^{*\top} \mu_{T+1}\|$$

$$= -\langle \frac{\hat{W}_1^\top \hat{\mu}_{T+1}}{\|\hat{W}_1^\top \hat{\mu}_{T+1}\|}, \hat{W}_1^\top \mu_{T+1} \rangle + \langle \frac{W_1^{*\top} \hat{\mu}_{T+1}}{\|W_1^{*\top} \hat{\mu}_{T+1}\|}, W_1^{*\top} \mu_{T+1} \rangle$$

$$- \langle \frac{W_1^{*\top} \hat{\mu}_{T+1}}{\|W_1^{*\top} \hat{\mu}_{T+1}\|}, W_1^{*\top} \mu_{T+1} \rangle + \|W_1^{*\top} \mu_{T+1}\|$$

$$\lesssim \|\hat{W}_1 - W_1^*\| \|\mu_{T+1}\| + \|W_1^{*\top} \mu_{T+1} - W_1^{*\top} \hat{\mu}_{T+1}\|$$

$$\lesssim \|\hat{W}_1 - W_1^*\| \|\mu_{T+1}\| + \|B^\top \mu_{T+1} - B^\top \hat{\mu}_{T+1}\|$$

if $n > r^2 \max\{p/T, \log(1/\delta)/T, 1\}$. The last formula is due to the fact that $W_1^*$ and $B$ are different only up to an orthogonal matrix.

By standard chaining techniques, we have with probability $1 - \delta$

$$\|B_1^\top \mu_{T+1} - B_1^\top \hat{\mu}_{T+1}\| \lesssim \sqrt{\frac{r}{n_{T+1}}} + \sqrt{\frac{\log(1/\delta)}{n_{T+1}}}.$$

Thus, we can further bound $\|\hat{W}_1 - W_1^*\|$ by $\sqrt{2} \|\sin \Theta(\hat{W}_1, B)\|_F$, thus, by Lemma 1, we have

$$\mathcal{R}(\hat{W}_1, \hat{w}_2^{(T+1)}) \lesssim \sqrt{\frac{r + \log(1/\delta)}{n_{T+1}}} + r(\sqrt{\frac{1}{n}} + \sqrt{\frac{p}{nT}} + \sqrt{\frac{\log(1/\delta)}{nT}}).$$

Now, if we further have $n > r n_{T+1}$, we have

$$\mathcal{R}(\hat{W}_1, \hat{w}_2^{(T+1)}) \lesssim \sqrt{\frac{r + \log(1/\delta)}{n_{T+1}}} + \sqrt{\frac{r^2 p}{nT}}.$$

$\square$

Plugging into $\delta = n^{-100}$, the proof is complete.

## C  Proof of Theorem 1

**Theorem 5 (Restatement of Theorem 1).** *Under Assumption 2 and 3, for $\|a_{T+1}\| = \alpha = \Omega(1)$, if $n > c_1 \max\{r^2, r/\alpha_T\} \cdot \max\{p \log T, \log n/T, 1\}$ and $n > c_2 (\alpha \alpha_T)^2 r n_{T+1}$ for universal constants $c_1, c_2$, $2r \le \min\{p, T\}$. There exists a universal constant $c_3$, such that if we choose $\varepsilon \in [\max_{t \in S_1} \|a_t\| + c_3 \sqrt{p \log T/n}, \min_{t \in S_2} \|a_t\| - c_3 \sqrt{p \log T/n}]$ (this set will not be empty if $T, n$ are large enough), for $\hat{W}_1^{adv}, \hat{w}_2^{adv,(T+1)}$ obtained in Algorithm 2 with $q = 2$, with probability at least $1 - O(n^{-100})$,*

$$\|\sin \Theta(\hat{W}_1^{adv}, B)\|_F \lesssim (\alpha_T)^{-1} \left( \sqrt{\frac{r^2}{n}} + \sqrt{\frac{pr^2}{nT}} + \sqrt{\frac{r^2 \log n}{nT}} \right),$$

*and the excess risk*

$$\mathcal{R}(\hat{W}_1^{adv}, \hat{w}_2^{adv,(T+1)}) \lesssim \alpha \sqrt{\frac{r + \log n}{n_{T+1}}} + (\alpha_T)^{-1} \left( \sqrt{\frac{r^2 p}{nT}} \right).$$

*Proof.* For $\ell_2$-adversarial training, we have

$$\hat{\beta}_t^{adv} = \text{argmin}_{\|\beta_t\|\leq 1} \max_{\|\delta_i\|_p \leq \varepsilon} \frac{1}{n_t} \sum_{i=1}^{n_t} -y_i^{(t)} \langle \beta_t, x_i^{(t)} + \delta_i \rangle$$

$$= \text{argmin}_{\|\beta_t\|\leq 1} \max_{\|\delta_i\|_p \leq \varepsilon} \frac{1}{n_t} \sum_{i=1}^{n_t} -y_i^{(t)} \langle \beta_t, x_i^{(t)} \rangle + \varepsilon\|\beta_t\|$$

Recall $\hat{\mu}_t = \frac{1}{n_t} \sum_{i=1}^{n_t} y_i^{(t)} x_i^{(t)}$, if we have $\|\hat{\mu}_t\| \geq \varepsilon$, then $\hat{\beta}_t^{adv} = \hat{\mu}_t/\|\hat{\mu}_t\|$, otherwise, $\hat{\beta}_t^{adv} = 0$.

We denote

$$\hat{G} = [\hat{\beta}_1^{adv}, \cdots, \hat{\beta}_T^{adv}].$$

Since $|S_1| = \Theta(T)$, there exists a universal constant $c_3$ such that with probability $1 - \delta$, we have for all $i \in S_1$, $\hat{\mu}_i \leq \|a_i\| + c_3 \sqrt{p \log T/n}$. Thus, if $T$ is large enough, the set $[\max_{t \in S_1} \|a_t\| + c_3\sqrt{p \log T/n}, \min_{t \in S_2} \|a_t\| - c_3\sqrt{p \log T/n}]$ is non-empty. If we choose $\varepsilon \in [\max_{t \in S_1} \|a_t\| + c_3\sqrt{p \log T/n}, \min_{t \in S_2} \|a_t\| - c_3\sqrt{p \log T/n}]$, for all $t \in S_2$, $\hat{\beta}_t^{adv} = \hat{\mu}_t/\|\hat{\mu}_t\|$. Meanwhile, $\hat{G}_{S_1}$ is a zero matrix.

Notice that the left singular vectors obtained by applying SVD to $\hat{G}$ for left singular vectors is equivalent to applying SVD for left singular vectors to $\hat{G}_{S_2}$, which is further equivalent to applying SVD for left singular vectors to $\hat{\Phi}_{S_2}$, given that $\hat{G}_2$ is equal to $\hat{\Phi}_{S_2}$ times a diagonal matrix on the right. Thus, we have

$$\|\sin\Theta(\hat{W}_1^{adv}, B)\|_F \lesssim \frac{(2\sigma_1(\Phi_{S_2}) + \|\hat{\Phi}_{S_2} - \Phi_{S_2}\|_{op}) \min\{r^{0.5}\|\hat{\Phi}_{S_2} - \Phi_{S_2}\|_{op}, \|\hat{\Phi}_{S_2} - \Phi_{S_2}\|_F\}}{\sigma_r^2(\Phi_{S_2})}.$$

By our assumptions, we know that

$$\mathbb{P}\left(\sup_{v \in \mathbb{S}^{p-1}} \|v^\top(\hat{\Phi}_{S_2} - \Phi_{S_2})\| \gtrsim \alpha_T^{-1}(\sqrt{\frac{p}{n}} + \sqrt{\frac{T}{n}} + \sqrt{\frac{\log(1/\delta)}{n}})\right) \leq \delta.$$

As a result,

$$\|\sin\Theta(\hat{W}_1, B)\|_F \lesssim \alpha_T^{-2} r\sqrt{r}(\frac{1}{n} + \frac{p}{nT} + \frac{\log(1/\delta)}{nT}) + \alpha_T^{-1} r(\sqrt{\frac{1}{n}} + \sqrt{\frac{p}{nT}} + \sqrt{\frac{\log(1/\delta)}{nT}}).$$

If we further have $n > \frac{r}{\alpha_T}\max\{p/T, \log(1/\delta)/T, 1\}$, we further have

$$\|\sin\Theta(\hat{W}_1, B)\|_F \lesssim (\alpha_T)^{-1} r(\sqrt{\frac{1}{n}} + \sqrt{\frac{p}{nT}} + \sqrt{\frac{\log(1/\delta)}{nT}}).$$

Now, if we further have $n > (\alpha\alpha_T)^2 r n_{T+1}$, we have

$$\mathcal{R}(\hat{W}_1, \hat{w}_2^{(T+1)}) \lesssim \alpha\sqrt{\frac{r + \log(1/\delta)}{n_{T+1}}} + (\alpha_T)^{-1}\sqrt{\frac{r^2 p}{nT}}.$$

Plugging into $\delta = n^{-100}$, the proof is complete.

$\square$

**Remark 6** ($\ell_2$-adversarial training v.s. standard training). *The proof of the counterpart of Lemma 1 under the setting of Theorem 1 basically folllows similar methods in the proof of Lemma 1. The only modification is that we need an extra step:*

$$\mathbb{P}\left(\sup_{v \in \mathbb{S}^{p-1}} \|v^\top(\hat{\Phi} - \Phi)\| \gtrsim \sqrt{\frac{p}{n}} + \sqrt{\frac{T}{n}} + \sqrt{\frac{\log(1/\delta)}{n}}\right) \leq \mathbb{P}\left(\sup_{v \in \mathbb{S}^{p-1}} \|v^\top(\hat{\Phi}_{S_1} - \Phi_{S_1})\| \gtrsim \sqrt{\frac{p}{n}} + \sqrt{\frac{T}{n}} + \sqrt{\frac{\log(1/\delta)}{n}}\right)$$

$$+ \mathbb{P}\left(\sup_{v \in \mathbb{S}^{p-1}} \|v^\top(\hat{\Phi}_{S_2} - \Phi_{S_2})\| \gtrsim \sqrt{\frac{p}{n}} + \sqrt{\frac{T}{n}} + \sqrt{\frac{\log(1/\delta)}{n}}\right)$$

*and recall that both $|S_1|$ and $|S_2|$ are of order $\Theta(T)$.*

# D  Proof of Theorem 2

**Theorem 6** (**Restatement of Theorem 2**). *Under Assumptions 1 and 4, if $n > c_1 \cdot r^2 \max\{s^2 \log^2 T/T, rn_{T+1}, 1\}$ for some universal constants $c_1 > 0$, $2r \leq \min\{p, T\}$. There exists a universal constant $c_2$, such that if we choose $\varepsilon > c_2 \sqrt{\log p/n}$, for and $\hat{W}_1^{adv}$, $\hat{w}_2^{adv,(T+1)}$ obtained in Algorithm 2 with $q = \infty$, with probability at least $1 - O(n^{-100}) - O(T^{-100})$,*

$$\|\sin\Theta(\hat{W}_1^{adv}, B)\|_F \lesssim r\left(\sqrt{\frac{1}{n}} + \sqrt{\frac{s^2}{nT}}\right) \cdot \log(T+p),$$

*and the excess risk*

$$\mathcal{R}(\hat{W}_1^{adv}, \hat{w}_2^{adv,(T+1)}) \lesssim \left(\sqrt{\frac{r + \log n}{n_{T+1}}} + r\sqrt{\frac{s^2}{nT}}\right) \cdot \log(T+p). \tag{7}$$

*Proof.* For $\ell_\infty$-adversarial training, we have

$$\hat{\beta}_t^{adv} = \mathrm{argmin}_{\|\beta_t\| \leq 1} \max_{\|\delta_i\|_\infty \leq \varepsilon} \frac{1}{n_t} \sum_{i=1}^{n_t} -y_i^{(t)}\langle \beta_t, x_i^{(t)} + \delta_i \rangle$$

$$= \mathrm{argmin}_{\|\beta_t\| \leq 1} \frac{1}{n_t} \sum_{i=1}^{n_t} -y_i^{(t)}\langle \beta_t, x_i^{(t)} \rangle + \varepsilon\|\beta_t\|_1$$

$$= \mathrm{argmin}_{\|\beta_t\| \leq 1} \langle \beta_t, \frac{1}{n_t} \sum_{i=1}^{n_t} -y_i^{(t)} x_i^{(t)} \rangle + \varepsilon\|\beta_t\|_1$$

Recall $\hat{\mu}_t = \frac{1}{n_t}\sum_{i=1}^{n_t} y_i^{(t)} x_i^{(t)}$. By observation, when reaching minimum, we have to have $sgn(\beta_{tj}) = sgn(\hat{\mu}_{tj})$, therefore

$$\mathrm{argmax}_{\|\beta_t\| = 1} \sum_{j=1}^{d} \hat{\mu}_{tj}\beta_{tj} - \varepsilon|\beta_{tj}|$$

$$= \mathrm{argmax}_{\|\beta_t\| = 1} \sum_{j=1}^{d} (\hat{\mu}_{tj} - \varepsilon \cdot sgn(\hat{\mu}_{tj}))\beta_{tj}$$

$$= \frac{T_\varepsilon(\hat{\mu})}{\|T_\varepsilon(\hat{\mu})\|},$$

where $T_\varepsilon(\hat{\mu})$ is the hard-thresholding operator with $(T_\varepsilon(\hat{\mu}))_j = sgn(\hat{\mu}_j) \cdot \max\{|\hat{\mu}_j| - \varepsilon, 0\}$.

We denote

$$\hat{G} = [\hat{\beta}_1^{adv}, \cdots, \hat{\beta}_T^{adv}].$$

By the choice of $\varepsilon$, $\varepsilon \gtrsim C\sqrt{\frac{\log p}{n}}$ for sufficiently large $C$, we have that the column sparsities of $\hat{G}$ is no larger than $s \log T$. As a result, the total number of non-zero elements in $\hat{G}$ is less than $O(Ts \log T)$ with probability at least $1 - T^{-100}$.

Now we divide the rows of $\hat{G}$ by two parts: $[p] = A_1 \cup A_2$, where $A_1$ consists of indices of rows whose sparsity smaller than or equal to $s$, and $A_2$ consists of indices of rows whose sparsity larger than $s$.

Since the number of non-zero elements in $\hat{G}$ is less than $Ts \log T$, we have $|A_2| \leq T \log T$. Using the similar analysis as in the proof of Lemma 1, we have

$$\|\hat{\Phi}_{A_2} - \Phi_{A_2}\| \leq \sqrt{\frac{T \log T}{n}}.$$

For the rows in $A_1$, all of them has sparsity $\lesssim s$, so the maximum $\ell_1$ norm of these rows

$$\|\hat{\Phi}_{A_1} - \Phi_{A_1}\|_\infty = O_P(s\sqrt{\frac{\log T}{n}}).$$

Similarly, the maximum $\ell_1$ norm of the columns in $\hat{G}_{A_1}$ satisfies

$$\|\hat{\Phi}_{A_1} - \Phi_{A_1}\|_1 = O_P(s\sqrt{\frac{\log p}{n}}).$$

Therefore, we have

$$\|\hat{\Phi}_{A_1} - \Phi_{A_1}\| \leq \sqrt{\|\hat{\Phi}_{A_1} - \Phi_{A_1}^*\|_\infty \|\hat{\Phi}_{A_1} - \Phi_{A_1}\|_1} = O_P(s\sqrt{\frac{\log p + \log T}{n}}).$$

Consequently,

$$\|\hat{\Phi} - \Phi\| \leq \|\hat{\Phi}_{A_1} - \Phi_{A_1}\| + \|\hat{\Phi}_{A_2} - \Phi_{A_2}\| = O_P(s\sqrt{\frac{\log p + \log T}{n}})$$

As a result, when $s\sqrt{\frac{\log p + \log T}{n}} \lesssim T/r$, applying Lemma 2, we obtain

$$\|\sin\Theta(\hat{W}_1, B)\|_F \lesssim \sin\theta(\hat{W}_1^{adv}, B) \lesssim (\sqrt{\frac{r}{n}} + \sqrt{\frac{rs^2}{nT}}) \cdot \log(T + p).$$

Now, if we further have $n > (\alpha\alpha_T)^2 n_{T+1}/\nu$, we have

$$\mathcal{R}(\hat{W}_1, \hat{w}_2^{(T+1)}) \lesssim \sqrt{\frac{r + \log(1/\delta)}{n_{T+1}}} + \sqrt{\frac{rs^2}{nT}} \cdot \log(T + p).$$

$\square$

**Remark 7** ($\ell_\infty$-adversarial training v.s. standard training). *The proof of the counterpart of Lemma 1 under the setting of Theorem 2 follows exact the same method in the proof of Lemma 1.*

## E  Proof of the case with pseudo-labeling

**Theorem 7** (Restatement of Theorem 3). *Denote* $\tilde{n} = \min_{t\in[T]} n_t^u$ *and assume* $\tilde{n} > c_1 \max\{pr^2/T, r^2\log(1/\delta)/T, r^2, n\}$ *for some constant* $c_1 > 0$. *Assume* $\sigma_r(M^\top M/T) = \Omega(1/r)$ *and* $n^{c_2} \gtrsim \tilde{n} \gtrsim n$ *for some* $c_2 > 1$, *if* $n \gtrsim (T + d)$ *and* $\min_{t\in[T]} \|a_t\| = \Theta(\log^2 n)$ *and* $\eta_i^{(t)} \sim \mathcal{N}_p(0, \rho_t^2 I^2)$ *for* $\rho_t = \Theta(1)$. *Let* $\hat{W}_{1,aug}$ *obtained in Algorithm 3, with probability* $1 - O(n^{-100})$,

$$\|\sin\Theta(\hat{W}_{1,aug}, B)\|_F \lesssim r\left(\sqrt{\frac{1}{\tilde{n}}} + \sqrt{\frac{p}{\tilde{n}T}} + \sqrt{\frac{\log n}{\tilde{n}T}}\right).$$

*Proof.* Let us first analyze the performance of pseudo-labeling algorithm in each individual task. In the following, we analyze the properties of $y_i^{u,(t)}$ and $\hat{\mu}_{final}^{(t)} = \frac{1}{n_t^u + n_t}\sum_{i=1}^{n_t^u + n_t}(\sum_{i=1}^{n_t^t} x_i^u y_i^u + \sum_{i=1}^{n_t} x_i^u y_i^u)$. Since $\tilde{n} \gtrsim n$ and we only care about the rate in the result. In the following, we derive the results for $\hat{\mu}_{final}^{(t)} = \frac{1}{n_t^u}\sum_{i=1}^{n_t^u + n_t}(\sum_{i=1}^{n_u^t} x_i^u y_i^u)$. Also, for the notational simplicity, we omit the index $t$ in the following analysis.

We follow the similar analysis of Carmon et al. [11] to study the property of $y_i^u$. Let $b_i$ be the indicator that the $i$-th pseudo-label is incorrect, so that $x_i^u \sim N((1-2b_i)y_i^u\mu, I) := (1-2b_i)y_i^u\mu + \varepsilon_i^u$. Then we can write

$$\hat{\mu}_{final} = \gamma\mu + \tilde{\delta},$$

where $\gamma = \frac{1}{n_u}\sum_{i=1}^{n_u}(1-2b_i)$ and $\tilde{\delta} = \frac{1}{n_u}\sum_{i=1}^{n_u}\varepsilon_i^u y_i^u$.

Let's write $y_i^u = sign(x_i^\top\hat{\mu})$. Using the rotational invariance of Gaussian, without loss of generality, we choose the coordinate system where the first coordinate is in the direction of $\hat{\mu}$. Then $y_i^u = sign(x_i^\top\hat{\mu}) = sign(x_{i1}) = sign(y_i^* \frac{\mu^\top\hat{\mu}}{\|\hat{\mu}\|} + \varepsilon_{i1}^u)$ and are independent with $\varepsilon_{ij}^u$ ($j \geq 2$).

As a result,

$$\frac{1}{n_u}\sum_{i=1}^{n_u}\varepsilon_{ij}^u \cdot y_i^u \stackrel{d}{=} \frac{1}{n_u}\sum_{i=1}^{n_u}\varepsilon_{ij}^u, \quad \text{for } j \geq 2.$$

Now let's focus on $\frac{1}{n_u}\sum_{i=1}^{n_u}\varepsilon_{i1}^u \cdot y_i^u$. Let $y_i^* = (1-2b_i)y_i^u$, we have

$$\frac{1}{n_u}\sum_{i=1}^{n_u}\varepsilon_{i1}^u \cdot y_i^u = \frac{1}{n_u}\sum_{i=1}^{n_u}\varepsilon_{i1}^u \cdot y_i^* + 2\frac{1}{n_u}\sum_{i=1}^{n_u}\varepsilon_{i1}^u \cdot b_i \stackrel{d}{=} \frac{1}{n_u}\sum_{i=1}^{n_u}\varepsilon_{i1}^u + 2\frac{1}{n_u}\sum_{i=1}^{n_u}\varepsilon_{i1}^u \cdot b_i.$$

Since

$$(\frac{1}{n_u}\sum_{i=1}^{n_u}\varepsilon_{i1}^u \cdot b_i)^2 \leq (\frac{1}{n_u}\sum_{i=1}^{n_u}(\varepsilon_{i1}^u)^2)(\frac{1}{n_u}\sum_{i=1}^{n_u}b_i^2) \lesssim \frac{1}{n_u}\sum_{i=1}^{n_u}b_i^2 = \frac{1}{n_u}\sum_{i=1}^{n_u}b_i \lesssim \mathbb{E}[b_i] + \frac{1}{\sqrt{n_u}} \lesssim +\frac{1}{n}+\frac{1}{\sqrt{n_u}},$$

where the last inequality is due to the fact that

$$\mathbb{E}[b_i] = \mathbb{P}(y_i^u \neq y_i^*) = \mathbb{P}(sign(y_i^*\frac{\mu^\top\hat{\mu}}{\|\hat{\mu}\|} + \varepsilon_{i1}^u) \neq y_i^*)$$

$$\leq \mathbb{P}(sign(y_i^*\frac{\mu^\top\hat{\mu}}{\|\hat{\mu}\|} + \varepsilon_{i1}^u) \neq y_i^* \mid \frac{\mu^\top\hat{\mu}}{\|\hat{\mu}\|} > \frac{1}{2}\|\mu\|) + \mathbb{P}(\frac{\mu^\top\hat{\mu}}{\|\hat{\mu}\|} > \frac{1}{2}\|\mu\|)$$

$$\lesssim \exp^{-\|\mu\|/2} + \frac{1}{n^C}$$

As a result, we have

$$\tilde{\delta} \stackrel{d}{=} \frac{1}{n_u}\sum_{i=1}^{n_u}\varepsilon_i^u + e,$$

where $\|e\|_2 \lesssim \frac{1}{\sqrt{n_u}} + \frac{1}{n^C}$.

Additionally, we have $\gamma = \frac{1}{n_u}\sum_{i=1}^{n_u}(1-2b_i) = 1 - \frac{2}{n_u}\sum_{i=1}^{n_u}b_i = 1 - O(\frac{1}{\sqrt{n_u}} + \frac{1}{n^C})$.

As a result, for each $t \in [T]$, we have

$$\hat{\mu}_t = \mu_t + \frac{1}{n_u}\sum_{i=1}^{n_u}\varepsilon_i^u + e',$$

with $\|e'\|_2 \lesssim \frac{1}{\sqrt{n_u}} + \frac{1}{n^{C'}}$ being a negligible term.

Since $e'$ is negligible, we can then follow the same proof as those in Section A by considering $\tilde{\mu}_t = \mu_t + \frac{1}{n_u}\sum_{i=1}^{n_u}\varepsilon_i^u$ and obtain the desired results.

Similarly, due to the negligibility of $e'$, we can prove Theorem 4 by following the exact same techniques in Sections C and D. □

## F   Lower bound proof

**Proposition 2** (Restatement of Proposition 1). *Let us consider the parameter space* $\Xi = \{A \in \mathbb{R}^{p\times r}, B \in \mathbb{R}^{p\times r} : \sigma_r(A^\top A/T) \gtrsim 1, B^\top B = I_r\}$. *If* $nT \gtrsim rp$, *we then have*

$$\inf_{\hat{W}_1}\sup_{\Xi}\mathbb{E}\|\sin\Theta(B,\hat{W}_1)\|_F \gtrsim \sqrt{\frac{rp}{nT}}.$$

We first invoke the Fano's lemma.

**Lemma 4** ([54]). *Let* $M \geq 0$ *and* $\mu_0, \mu_1, ..., \mu_M \in \Theta$. *For some constants* $\alpha \in (0, 1/8), \gamma > 0$, *and any classifier* $\hat{G}$, *if* $\mathrm{KL}(\mathbb{P}_{\mu_i}, \mathbb{P}_{\mu_0}) \leq \alpha \log M$ *for all* $1 \leq i \leq M$, *and* $L(\mu_i, \mu_j)$ *for all* $0 \leq i \neq j \leq M$, *then*

$$\inf_{\hat{\mu}}\sup_{i\in[M]}\mathbb{E}_{\mu_i}[L(\mu_i, \hat{\mu})] \gtrsim \gamma.$$

Now we take $B_0, B_1, ..., B_M$ as the $\eta$-packing number of $O^{p \times r}$ with the $\sin \theta$ distance.

Then according to [41, 52], we have

$$\log M \asymp rd \log(\frac{1}{\eta}).$$

For any $i \in [M]$, we have

$$\mathrm{KL}(\mathbb{P}_{B_i}, \mathbb{P}_{B_0}) = \sum_{t=1}^{T} n\|(B_i - B_0)a_t\|^2 \leq nT\eta^2.$$

Let $\eta = \sqrt{\frac{rd}{nT}}$, we complete the proof.

## G   Additional Empirical Results

We provide additional results on transfer performance with varied amounts of pseudo-labels in Table 2. Here, we train models with both adversarial (allowed maximum perturbations of $\varepsilon = 1$ with respect to the $\ell_2$ norm) and non-adversarial (standard) training on ImageNet. The observed trend is the same as on the CIFAR-10 and CIFAR-100 tasks from Table 1 – both using robust training and additional pseudo-labeled data improve performance.

Table 2: Additional results extending Table 1. Effect of amount of pseudo-labels on transfer task performance (measured with accuracy). At $0\%$, we just use $10\%$ of data from the source task; at $900\%$, we use all remaining $90\%$ of data with pseudo-labels (this is 9 times the train set size). Adversarial training corresponds to using $\ell_2$-adversarial training with $\varepsilon = 1$ on the source task. As per Section 7 of [46], images in all datasetsare down-scaled to $32 \times 32$ before scaling back to $224 \times 224$.

| Source Task | Target Task | +0% Pseudo-labels | +200% Pseudo-labels | +500% Pseudo-labels | +900% Pseudo-labels |
|---|---|---|---|---|---|
| ImageNet | Aircraft [35] | 17.3% | 17.6% | 17.9% | 19.9% |
| ImageNet (w/adv.training) | Aircraft | 21.2% | 20.9% | 24.0% | 24.5% |
| ImageNet | Flowers [40] | 60.7% | 64.9% | 65.4% | 66.5% |
| ImageNet (w/adv.training) | Flowers | 66.9% | 68.1% | 70.0% | 70.1% |
| ImageNet | Food [8] | 33.7% | 36.0% | 36.7% | 37.2% |
| ImageNet (w/adv.training) | Food | 35.8% | 37.5% | 39.4% | 40.8% |
| ImageNet | Pets [42] | 43.2% | 44.9% | 48.4% | 49.0% |
| ImageNet (w/adv.training) | Pets | 47.9% | 53.1% | 58.9% | 59.6% |

## H   Experiment Details

### H.1   Training Hyperparameters

All of our experiments use the ResNet-18 architecture. When transferring to the target task, we only update the final layer of the model. Our hyperparameter choices are identical to those used in [46]:

1. ImageNet (source task) models are trained with SGD for 90 epochs with a momentum of $0.9$, weight decay of $1e - 4$, and a batch size of $512$. The initial learning rate is set to $0.1$ and is updated every 30 epochs by a factor of $0.1$. The adversarial examples for adversarial training are generated using 3 steps with step size $\frac{2\varepsilon}{3}$.

2. Target task models are trained for 150 epochs with SGD with a momentum of $0.9$, weight decay of $5e - 4$, and a batch size of $64$. The initial learning rate is set to $0.01$ and is updated every 50 epochs by a factor of $0.1$.

Data augmentation is also identical to the methods used in [46]. As per Section 7 of [46], we scale all our target task images down to size $32 \times 32$ before rescaling back to size $224 \times 224$.

Experiments were run on a GPU cluster. A variety of NVIDIA GPUs were used, as allocated by the cluster. Training time for each source task model was around 2 days (less when using subsampled data) using 4 GPUs. Training time for each target task model was typically between 1-5 hours (depending on the dataset) using 1 GPU.

## H.2  Pseudo-label Generation

When subsampling ImageNet (our source task), the sampled 10% with ground truth labels preserves the class label distribution. This sample is fixed for all our experiments. All ImageNet pseudo-labels are generated by a model trained on this 10% without any adversarial training. This model has a source task test accuracy (top-1) of $44.0\%$.

When training models with pseudo-labels, we preserve the class label distribution of the original training set (i.e., we add less pseudo-labels for those classes that have fewer examples in the entire training set).