# OpenReview forum: "Adversarial Training Helps Transfer Learning via Better Representations"
_NeurIPS.cc/2021/Conference — NeurIPS 2021 Poster_

### Official Review · Reviewer_bcKd · 2021-07-15

**Rating:** 5
**Confidence:** 2

**Summary:**

This paper reveals, both theoretically and empirically, that adversarial training is beneficial to transfer learning, which helps learning a more robust representation and therefore boosts the predictive performance on the target task. Additionally, pseudo-labeling is claimed to have complementary benefits and can further improve the transfer.

**Ethical Concerns:**

I think there are no ethical issues with this paper.



**Limitations And Societal Impact:**

Please refer to the section "Main Review".



**Main Review:**

While attempting to tackle an important problem, this paper, at least in its current form, has several flaws.
1. The experiments are not very solid. It would have been nice to see some experiments on another domain than image classification, to see how this method generalizes.
2. And, I would like to know, when adversarial training is directly plugged to the existing transfer learning method, in theory, under what conditions will it be effective?

**Time Spent Reviewing:**

6

---

> ### Author Response · Authors · 2021-08-10
> **Thank you for your reviews**
>
> Thanks for your comments and suggestions. We respond to your concerns point by point in the following:
>
>
> 1. Thanks for your suggestion of adding more experiments besides image classification.
> - First of all, we want to point out that there **might be some misunderstanding**: our paper is to initialize a rigorous theoretical investigation of how adversarial learning helps transfer learning instead of empirically investigating this problem. We want to note that the empirical experiments have been extensively done in the previous literature such as [1,2]. However, the previous works did not provide theoretical analysis of *how* adversarial training helps transfer learning, which is the main contribution of our work.
> - Based on your suggestion, we ran an additional experiment in bounding box prediction, which is a regression problem rather than classification. The task is to predict the bounding box coordinates of the traffic sign in the input image. We use the same ResNet model pretrained on ImageNet with varying levels of robustness, and transfer to this regression task, updating only the final layer predicting bounding box coordinates on a small dataset of 877 traffic sign images. The reported error is the $l_1$ loss summed across upper left corner and bottom right corner bounding box predictions, normalized to the image size (i.e., upper left corner of the image is (-1,-1) and the bottom right corner of the image is (+1,+1)). Smaller error is better here. Consistent with other results in our paper, models with moderate levels of robust training outperform the model with no robust training in this regression task. The transfer performance of the model with no robust training ($\epsilon = 0$) is 1.08 +/- 0.054 in $l_1$ test error. With $\epsilon = 0.1$ robust training, the transfer $l_1$ error drops to 1.03 +/- 0.049. And with $\epsilon = 1$ robust training, the transfer $l_1$ error drops to 1.00 +/- 0.041. Recall that $\epsilon$ is the maximum allowed $l_2$ perturbation size.
>
>
>
> 2. Our paper indeed studied the method “when adversarial training is directly plugged to the existing transfer learning method.” Our method closely follows an existing method of transfer learning, the so-called fixed-feature transfer learning [1]. In the fixed-feature transfer learning, one first obtains a representation trained from source tasks (the first few layers of a neural network), then for downstream tasks, one fixes the representation and only trains additional task-specific parts (for instance, the last few layers of NN). In [1], the authors found that directly plugging the adversarial training in the representation learning step yields better performance in downstream tasks, and **we exactly studied such a method in theory**, and demonstrated under which conditions the adversarial training will be effective. Our method closely follows the recent theoretical literature [4] on **standard transfer learning** and **modifies the representation learning part by plugging in adversarial training** and our paper is the first to study how adversarial training can help transfer learning in theory.
>
> We are thankful for your suggestions and hope that our response can ease your concerns, clarify the misunderstanding, and help the reviewer to re-evaluate our paper.
>
>
> ​​[1] Hadi Salman, Andrew Ilyas, Logan Engstrom, Ashish Kapoor, and Aleksander Madry.  Do adversarially robust imagenet models transfer better? arXiv preprint arXiv:2007.08489, 2020
>
> [2] Francisco Utrera, Evan Kravitz, N Benjamin Erichson, Rajiv Khanna, and Michael W Mahoney. Adversarially-trained deep nets transfer better: Illustration on image classification. ICLR 2020.
>
> [3] Simon S Du, Wei Hu, Sham M Kakade, Jason D Lee, and Qi Lei. Few-shot learning via learning the representation, provably. arXiv preprint arXiv:2002.09434, 2020.
>
> [4] Nilesh Tripuraneni, Chi Jin, and Michael I Jordan. Provable meta-learning of linear representations. arXiv preprint arXiv:2002.11684, 2020.

---

> > ### Author Response · Authors · 2021-08-19
> > **We would love to hear back from Reviewer bcKd**
> >
> > Dear Reviewer bcKd,
> >
> > We really appreciate your comment and would love to see whether you still have concerns about our paper. To recap,
> >
> > 1. We added more experiments according to your suggestions.
> >
> > 2.  We clarified that in our paper, we presented the technical conditions for the method “when adversarial training is directly plugged to the existing transfer learning method.”
> >
> > We are more than happy to further provide explanations or clarifications if more is needed. Thank you!

---

> > > ### Author Response · Authors · 2021-09-02
> > > **Would love to hear more from the reviewer**
> > >
> > > Dear reviewer, please feel free to tell us if you have any further concerns. Given that the discussion phase will end soon, we sincerely hope you could reevaluate our paper and raise our score if your concerns are addressed.

---

### Official Review · Reviewer_ZMUB · 2021-07-15

**Rating:** 6
**Confidence:** 2

**Summary:**

This work proposes a theoretical perspective on why adversarial training in the source domain improves the fixed-feature transfer learning ability. The authors further show both pseudo-labeling and adversarial training can lead to better representation.


**Ethics Review Area:**

["I don’t know"]

**Main Review:**

- Originality: The proposed theoretical analysis is the first analysis on the effect of adversarial training on transfer learning. This is quite novel and non-trivial to establish theoretical analysis covering few common setups in $l_2$, $l_
{\infty}$, and pseudo-labeling scenarios.

- Quality: The proposed technical framework seems general sounds (Note I didn’t examine the proof in details).

- Clarity:
  * Starting from page 4, I found it a bit hard to follow. Perhaps, it is due to the use of many symbols or without reading the proofs. From readers, it requires going back-and-forth to check the original definition and assumptions. It might be helpful to add some intuition behind the algorithm, theory, etc. I also found several descriptions are not easy to follow.
  For example,
    * (a) L167, “we consider the low-rank case” --> it is not immediately clear it is referring to M.
    * (b) L172, "the task-specific error is easy to deal with." --> it will be helpful to expand this in more detail.
    * (c) In L202-203, it is not obvious to me for the conclusion drawing from uniform signal-to-noise ratios.
    * (d) In L319-320, it is not clear to me how to lead to the conclusion in varying signal to noise ratio ($l_2$ adversarial training) or with many redundant features ($l_\infty$ adversarial training).

  * Few citation format could be improved. For example, in [43, 44, 56, 57], the transaction name should be Capitalized; [45] does not cover the full author; In [58] of supplementary material, davis–kahan -> Davis–Kahan. (only list a few of them here)
  * L112, add a space between "learning" and "[20, 47]".



- Significance: the proposed results seem sound and justify few previously empirical observations. I wonder whether it is possible to extend the framework to justify the improved adversarial robustness during transfer learning (even without adversarial training on the target domain), empirically shown in [1].

  [1] https://openreview.net/forum?id=bgQek2O63w





**Time Spent Reviewing:**

2

---

> ### Author Response · Authors · 2021-08-10
> **Thank you for reviewing our paper**
>
> We would like to thank the reviewer for confirming the novelty and non-triviality of our results and leaving insightful comments and suggestions. We respond to your comments point by point in the following:
>
> 1. Thank you for your careful reading. We will modify our paper accordingly. Specifically,
>
> - (a) For the sentence "we consider the low-rank case...", we have changed it to "Throughout the paper, we consider the matrix $M$ to be low rank, whose rank $r$ is much smaller than $T$ and $p$. "
>
> - (b) For the sentence "the task-specific error is easy to deal with”, we meant that by the sub-gaussianity of data points, and using standard concentration inequalities, we can directly obtain an upper bound for the task-specific error $\|B^T\hat\mu_{T+1}$ $-B^T\mu_{T+1}\|$. For more details about standard techniques for concentration bounds of sub-gaussian vectors, please refer to Appendix at line 573. In the revised version, we’ve changed the sentence to “the task-specific error is easy to deal with following the standard concentration inequalities, and we present the derivation and result in appendix B.”
>
> - (c) Sorry for the confusion. The “uniform signal-to-noise ratio” refers to the case where the scale of $\|a_t\|$’s are all the same among $t\in[T]$. As we point out in Proposition 1 and the discussion in line 202-207, if we directly study the case where the scale of $\|a_t\|$’s are all the same among $t\in[T]$ (such as those in the papers [1],[2]), Algorithm 1 without adversarial training already is rate-optimal in the sense of achieving the minimax optimal rate.
>
> - (d) In line 319-320, it is a specific illustration of the results obtained in Theorems 3 and 4. We can see the bounds are sharper via adversarial training and "the varying signal to noise ratio” corresponds to the Assumption 2 for $l_2$ adversarial training) or “with many redundant features” corresponds to Assumption 4 for $l_\infty$ adversarial training. We’ve added more explicit connections between this sentence and Assumptions 2 and 4 in the revised version.
>
> 2. Thank you for pointing the format issues out. We’ve corrected the citation format and the typos you mentioned.
>
> 3. Thanks for introducing the reference. It is interesting since the BYORL method proposed in the paper you brought up suggested obtaining a robust representation with self-supervised learning without labels (and even without adversarial training) can be further leveraged by a linear classifier to gain adversarial robustness. This is very interesting since annotated data is scarce in many cases. We will seriously consider how to extend our theory to this setting. Thanks for this suggestion!
>
>
>
> [1] Simon S Du, Wei Hu, Sham M Kakade, Jason D Lee, and Qi Lei. Few-shot learning via learning the representation, provably. ArXiv preprint arXiv:2002.09434, 2020.
>
> [2] Nilesh Tripuraneni, Chi Jin, and Michael I Jordan. Provable meta-learning of linear representations. ArXiv preprint arXiv:2002.11684, 2020.
>
> [3] Martin J Wainwright. High-dimensional statistics: A non-asymptotic viewpoint, volume 48. Cambridge University Press, 2019.

---

> > ### Author Response · Authors · 2021-08-19
> > **We would love to hear back from Reviewer ZMUB**
> >
> > Dear Reviewer ZMUB,
> >
> > We really appreciate your comment and would love to see whether you still have concerns about our paper. To recap,
> >
> > 1. We have rewritten the sentences according to your suggestions.
> >
> > 2.  We have also corrected the typos in the citation and added more discussions on the reference you mentioned.
> >
> > We are more than happy to further provide explanations or clarifications if more is needed. Thank you!

---

> > > ### Author Response · Authors · 2021-09-02
> > > **Would love to hear more from the reviewer**
> > >
> > > Dear reviewer, please feel free to tell us if you have any further concerns. Given that the discussion phase will end soon, we sincerely hope you could reevaluate our paper and raise our score if your concerns are addressed.

---

### Official Review · Reviewer_Wuja · 2021-07-16

**Rating:** 5
**Confidence:** 4

**Summary:**

The authors attempt to provide a theoretical explanation on why adversarial training helps with transfer of learned representations sometimes. The main deficiency is the lack of explanation and verification of the underlying assumptions of their analysis, making it hard to connect what they proposed to what we observe in practice.


**Limitations And Societal Impact:**

Yes

**Main Review:**

- The authors propose a theoretical explanation on why adversarial training help with transfer learning. It is an interesting attempt to explain why models trained on ImageNet transfer better to CIFAR with adversarial training.

- The main problem I find with the theoretical explanation offered in this paper is the validity of the assumptions. Assumption 2 is very strong: the paper is assuming unbounded signal-to-noise ratio difference among the tasks in terms of their regression vector a_t. This is very unusual. Or if we take the view from regression or optimization, these are a set of very badly scaled features or badly conditioned optimization problems.

- The second problem with this set of assumptions is that we cannot (or the authors have not) empirically verify them on actual data. So we cannot really make the connection that this signal-to-noise ratio difference problem is what makes adversarial training works in practice.

- There could be simpler explanations on why adversarial training helps transfer from Imagenet to CIFAR. CIFAR images are of much lower resolution compared to Imagenet, and adversarial training could help with making the model less reliant on low level filter features, which are dependent on resolutions. Of course this hypothesis also needs to be verified in some way (e.g., by checking the low level filters), but I am putting this up as an example that any explanation on why adversarial training helps transfer need to make a connection to the real data.

- The part on pseudo-labeling does not quite fit well with the rest of the paper. It is only introduced in the experiment section and was not considered as part of the analysis. It is nice to see the empirical results, but does not add to the coherence of the paper.

- Overall I think the issues with establishing the assumptions used in the theoretical analysis in this paper prevents me from recommending this paper for publication.


**Time Spent Reviewing:**

3

---

> ### Author Response · Authors · 2021-08-10
> **Thank you for reviewing our paper**
>
> Response: Thanks for your comments and suggestions. We respond to your concerns point by point in the following:
>
> 1. Regarding Assumption 2, we would like to explain why it's reasonable from three aspects. First, our paper is the first to provide the theoretical study of how adversarial training helps transfer learning by considering theoretical models that were used in the recent transfer learning literature [1,2]. We show that Assumption 2 is not only sufficient but also a **necessary** assumption to guarantee that adversarial training would significantly help transfer learning when $r$ is fixed. As we point out in Proposition 1 and the discussion in line 202-207, when Assumption 2 does not hold, that is, when the scale of $\|a_t\|$’s are all of the same constant order for $t\in[T]$ (such as those in the papers [1,2]), Algorithm 1 **without** adversarial training is already rate-optimal up to $\sqrt{r}$ in the sense of reaching the minimax-optimal rate. Therefore, Assumption 2 is necessary to ensure that adversarial training can achieve sharper convergence rates than the standard training. It provides interesting insights about when adversarial training can and cannot help transfer learning asymptotically. Second, in our proof, we provide a non-asymptotic analysis. Even if we only require a small portion of tasks with a large **constant** signal-to-noise ratio, we can still get smaller bounds with better constants via adversarial training, although the convergence rates are still of the same order. Third, in Assumption 2, intuitively speaking, we only need a small portion of tasks with a large signal-to-noise ratio when the number of tasks grows (not sample size), it is natural and is expected in the sense that there will be more tasks with badly scaled features when we have more tasks. We will add more discussions about all of these in the revision.
>
> 2. Thank you for your comment on connecting the theory to real data. We would like to point out that it is standard in machine learning and statistics literature to demonstrate noise-signal ratio by misclassification error, such as in [3]--- smaller misclassification error in practice implies larger noise-signal ratio. Therefore, one could empirically verify our assumption of signal-to-noise ratio on real data.
>
> 3. Thank you for providing your intuition. We discuss it in detail below.
> - In fact, your intuition is captured by our model. In the field of signal processing, it is a common technique to represent an image through wavelet basis expansion. After the wavelet transformation, the low-resolution image is a sparse vector in the transformed space, which corresponds to our model in Section 3.3. In our theory, we showed that adversarial training helps to promote such sparsity and enforce the model to ignore the redundant features (the redundant features here corresponds to the high-resolution part of the image in this example), which aligns with your intuition that “adversarial training could help with making the model less reliant on low-level filter features” (as low-level filters find the high-resolution part and high-level filters find of the low-resolution part the image).  Additionally, the analysis of our Theorem 1 suggests another intuition: when there are multiple source tasks, the adversarial training will not only make those tasks with small signal-to-noise ratios harder to perform classification (increase bias), but also make these tasks less competitive (decrease variance). Therefore, adversarial training will bias the model to focus on learning the representation out of those with large signal-to-noise ratios and improves the transferability.
>
> - We also conducted **two new experiments** to show that our results are not purely driven by resolution. First, we show that adversarial training helps transfer learning when we transfer between two datasets of similar low resolutions, SVHN and MNIST. We train a digit classification model on SVHN and transfer to MNIST. Images in SVHN are 32x32 and images in MNIST are 28x28 pixels; we pad each image and crop to get a 32x32 image for each dataset.  We follow the same setup as described in our paper, where we train a ResNet model on SVHN using either no robust training or varying levels of robust training (adversarial $l_2$ perturbations, with varying levels of maximum allowed $l_2$ perturbation size). Then we fix all layers in the network except the last layer and transfer to MNIST. We observe similar results as in our paper, where the models trained with moderate levels of robustness achieve higher accuracy. The model without adversarial training achieves transfer test accuracy of 82.5%. The models trained with adversarial training corresponding to $\epsilon = 0.05, 0.1, 0.25, 0.5$ achieve transfer performance of 84.2%, 89.3%, 93.1% and 89.9%, respectively. Recall that $\epsilon$ is the maximum allowed $l_2$ perturbation size.
>
> - In our second experiment, we transfer between ImageNet and the traffic sign dataset which has similar resolution as ImageNet (224x224 pixels). The task is to predict the bounding box coordinates of the traffic sign in the input image. We use the same ResNet model pretrained on ImageNet with varying levels of robustness, and transfer to this regression task, updating only the final layer predicting bounding box coordinates on a small dataset of 877 traffic sign images. The reported error is the $l_1$ loss summed across upper left corner and bottom right corner bounding box predictions, normalized to the image size (i.e., upper left corner of the image is (-1,-1) and the bottom right corner of the image is (+1,+1)). Smaller error is better here. Consistent with other results in our paper, models with moderate levels of robust training outperform the model with no robust training. The transfer performance of the model with no robust training ($\epsilon = 0$) is 1.08 +/- 0.054 in $l_1$ test error. With $\epsilon = 0.1$ robust training, the transfer $l_1$ error drops to 1.03 +/- 0.049. And with $\epsilon = 1$ robust training, the transfer $l_1$ error drops to 1.00 +/- 0.041.
>
> 4. We are sorry for the misunderstanding here. However, we indeed **complement our pseudo-labeling experiments with theoretical analysis** (Theorems 3 and 4 in Line 303 and 310) to illustrate how using pseudo-labeling in the source data can improve transfer learning and how adversarial training can further boost that improvement in theory.
>
> Thank you for your helpful suggestions and we hope that our response clarifies your questions. Our paper provides one of the first theoretical understanding of both necessary and sufficient conditions for adversarial training to improve transfer learning. We also show both empirically and theoretically how semi-supervised learning can similarly improve representation learning for transfer. We would really appreciate it if you can re-evaluate our paper given our response. Thank you!
>
> [1] Simon S Du, Wei Hu, Sham M Kakade, Jason D Lee, and Qi Lei. Few-shot learning via learning the representation, provably.
> arXiv preprint arXiv:2002.09434, 2020.
>
> [2] Nilesh Tripuraneni, Chi Jin, and Michael I Jordan. Provable meta-learning of linear representations. arXiv preprint arXiv:2002.11684, 2020.
>
> [3] Martin J Wainwright. High-dimensional statistics: A non-asymptotic viewpoint, volume 48. Cambridge University Press, 2019.

---

> > ### Author Response · Authors · 2021-08-19
> > **We would love to hear back from Reviewer Wuja**
> >
> > Dear Reviewer Wuja,
> >
> > We really appreciate your comment and would love to see whether you still have concerns about our paper. To recap,
> > 1. We showed that our Assumption 2 is not only a sufficient but also a necessary assumption to guarantee that adversarial training would significantly help transfer learning.
> >
> > 2. We added more discussion and experiments regarding the connection between our technical results and real data analysis.
> >
> > 3. We clarified that we have already complemented our pseudo-labeling experiments with theoretical analysis.
> >
> > We are more than happy to further provide explanations or clarifications if more is needed. Thank you!

---

> > > ### Comment · Reviewer_Wuja · 2021-08-27
> > > **Thank you for the response**
> > >
> > > Thanks for the detailed response. Two quick quesitons:
> > > 1. Where is the improved constant for non-asymptotic analysis in the appendix?
> > > 2. The analysis is based on having a large number of tasks, but in practice we are only transferring from one (or a few) tasks to another. It seems there is still a fairly big gap between theory and experiments

---

> > > > ### Author Response · Authors · 2021-08-28
> > > > **Thank you for your additional comments**
> > > >
> > > > Dear Reviewer Wuja,
> > > >
> > > > Thank you for your further feedback and we will answer your two questions one by one in the following.
> > > >
> > > > 1. To see the improvement in constant, let us look into the following derivations. First, according to the proof in line 579-599 in the appendix,  we can observe that as long as  the norms of $a_t$'s for $t\in S_2$ satisfy $\min_{t\in S_2}||a_t||-c\sqrt{p\log T/n}> \max_{t\in S_1}||a_t||+c\sqrt{p\log T/n}$ for some universal constant $c$, our proof would still work.  Since the norms of $a_t$'s for $t\in S_1$ are of constant orders in our setting, this essentially implies that as long as the norms of $a_t$'s for $t\in S_2$ are larger than some large enough constant $C_a>0$, our proof would still work. Therefore, following the proof in line 579-599, we can obtain, for $\ell_2$-adversarial training,
> > > > $$||\sin\Theta(\hat{W}^{adv}, B)||_F\le c' C_a^{-1}(\sqrt{r^2/n}+\sqrt{pr^2/(n T)}+\sqrt{r^2\log n/(n T)}),$$
> > > > for some constant $c'$ that does not depend on $C_a$ (here we provide a more detailed non-asymptotic description than the proof in our appendix in order to show the improvement in constant).  Additionally, according to the proof of Lemma $1$ in line 560-562 in the appendix, we have the following result for the standard training:
> > > > $$||\sin\Theta(\hat{W}, B)||_F\le c'' (\sqrt{r^2/n}+\sqrt{pr^2/(n T)}+\sqrt{r^2\log n/(n T)}),$$
> > > > where the constant $c''$ does not depend on $C_a$. Therefore, comparing the above two inequalities, we can see that when $C_a$ is large enough, we will have improved constant by using $\ell_2$-adversarial training.
> > > >
> > > > 2. We want to clarify that all of our theory applies when the number of source tasks is small (e.g. 3 or 4). Our analysis is non-asymptotic and explicitly shows how the number of tasks affects the performance of transfer learning. While we used some assumptions to rigorously prove our guarantees, recent transfer learning theory papers make similar assumptions (e.g. Tripuraneni et al. ICML 2021, and Du et al. ICLR 2021). Our paper is the *first work* to mathematically show how adversarial training helps representation learning for transfers, which is recognized as an important open problem for the ML theory community (e.g. stated in Tripuraneni et al. ICML 2021). We agree with the reviewer that there are rooms to extend the theory, and our work makes useful contributions by building the critical first foundations.
> > > >
> > > >
> > > >
> > > > [1] Nilesh Tripuraneni, Chi Jin, and Michael I Jordan. Provable meta-learning of linear representations. ICML 2021.
> > > >
> > > > [2] Simon S Du, Wei Hu, Sham M Kakade, Jason D Lee, and Qi Lei. Few-shot learning via learning the representation, provably. ICLR 2021.

---

> > > > > ### Author Response · Authors · 2021-09-02
> > > > > **Would love to hear more from the reviewer**
> > > > >
> > > > > Dear reviewer, please feel free to tell us if you have any further concerns. We hope that our answers regarding the gap of our theory and application, as well as the improvement of constant, have eased your concerns.
> > > > >
> > > > > Given that the discussion phase will end soon, we sincerely hope you could reevaluate our paper and raise our score if your concerns are addressed.

---

### Decision · Program_Chairs · 2021-09-27

**Decision:**

Accept (Poster)

**Comment:**

This is a solid paper that studies an important phenomenon that is not well understood: why adversarial training on the source data can improve the ability of models to transfer to new domains or targets? This is somewhat surprising as it might be easier to think of adversarial training as a task specific modification that can reduce performance on the source data, and the connection to transfer learning is not very intuitive on a first sight.

The paper replicates an adversarially trained ($\ell_2$ robustness) Imegenet model providing better accuracy on CIFAR10. Starting from fixed-feature transfer learning (Salman et.al.), an adversarial training step is plugged into the fixed-feature learning step.The authors also show that pseudo-labeling can lead to better representations. The authors analyse a hierarchical linear model for multitask learning and provide Alg-1 as the linear representation learning algorithm. This setup gives an upper bound to excess risk that can be decomposed in two terms: representation error (related to the sum of angles between the estimated and true factor matrices) and a task dependent term. The representation error can be further bounded using a Davis-Kahan style perturbation bound (Wedin's theorem). The strategy of the paper is to extend Alg-1 to the adversarial training setting (Alg-2) and finally also to include pseudo labelling (Alg-3).

The structure and the arguments of the paper are easy to follow. During rebuttal, several concerns were raised, I summarize main ones succinctly and less nuanced below:

1. The validity of the assumptions. Assumption 2 is very strong. No verification of assumptions. (This assumption is related to the magnitude of task specific weights $\alpha_k$ where a large signal-to-noise (SNR) corresponds to a large $||\alpha||$, a proxy of task uncertainty)

2. Pseudolabel scenario is not much related

3. Observed robustness could be explained by different resolution on the source and target datasets, Analysis does not explain/has a big gap to experiment as it requires too many tasks

In my opinion, the authors very clearly addressed the concerns raised by the reviewers. In particular, the connection of variation in the
SNR's to explain why adversarial training might help seems to come out naturally from the analysis and seems to be quite interesting.

In particular, I feel the paper makes some relevant contributions.

It carefully builds and contrasts the various algorithms and provides a compelling explanation for the observed phenomenon. Adversarial training will bias the model to focus on learning the representation out of those with large signal-to-noise ratios and enhances  transferability.

In my opinion, one possible useful addition to address point 3 above would have been to verify the theory with a set of synthetic simulations. This would be fairly straightforward starting from the model and could potentially illustrate the assumptions much more clearly by explicitly controlling the task distribution. Especially, I can envision an experiment that directly controls the task diversity and SNR's to empirically show the regime where robust training is helping. Many related concerns (regarding non-asymptotic validity) could have been partially addressed through such a synthetic experiment.

Overall, I am persuaded that the work would be a solid contribution and in the lack of strong counter arguments from the reviewers, I am using my own judgement and independent evaluation to suggest acceptance.